# Colorimetric identification of colorless acid vapors using a metal-organic framework-based sensor

Wonhyeong Jang [1,2], Hyejin Yoo [1,2], Dongjun Shin[1], Seokjin Noh[1] & Jin Yeong Kim [1] ✉

In terms of safety and emergency response, identifying hazardous gaseous acid chemicals is crucial for ensuring effective evacuation and administering proper first aid. However, current studies struggle to distinguish between different acid vapors and remain in the early stages of development. In this study, we propose an on-site monitorable acid vapor decoder, MOF-808-EDTA-Cu, integrating the robust MOF-808 with Cu-EDTA, functioning as a proton-triggered colorimetric decoder that translates the anionic components of corrosive acids into visible colors. The sensor exhibits a cyan-to-yellow shift when exposed to HCl vapor and can visually differentiate various acidic vapors (HF, HBr, and HI) through unique color changes. Furthermore, the compatibility of the MOF-based sensor with multiple metal ions having atomic-level dispersion broadens its discrimination range, enabling the identification of six different colorless acid vapors within a single sensor domain. Additionally, by incorporating a flexible polymer, the MOF-808-EDTA-Cu has been successfully processed into a portable miniaturized acid sensor, exhibiting distinct color changes that can be easily monitored by the naked eye and camera sensors. This provides experimental validation as a practical sensor capable of on-site 24-hour monitoring in the real world.

Acids are essential reagents in modern chemical processes and are used to produce various chemicals, such as fertilizers, detergents, batteries, and pharmaceuticals, as well as to remove impurities from many products[1–6]. Despite their versatility, acids pose significant risks owing to their corrosive and chemically reactive nature, which can cause irritation or severe damage to the eyes, skin, and respiratory system upon exposure to high concentrations or prolonged exposure[7,8]. The required emergency response and first-aid protocols depend on the type of acid exposed[9,10]; hence, the detection and identification of hazardous acid chemicals is of paramount importance for applications ranging from health diagnostics to public safety and environmental protection[1,2,11,12]. Electrochemical acid vapor sensors, some of which are already commercially available, exhibit high sensitivity, however, their acid vapors identification efficiency is limited[13,14].

Moreover, traditional acid vapor identification methods, including infrared and mass spectrometry, require complex and expensive instrumentation, rendering on-site identification of acid vapors challenging[15–17].

One of the most attractive approaches in this field involves the construction of colorimetric molecular decoders that exhibit acid-ochromism and vapochromism, offering simplicity in identifying acid vapors with the naked eye, cost efficiency, and the potential for on-site identification of multiple targets. Recently, several colorimetric sensors for the detection of acid vapors have been reported utilizing organic dyes[18–21], polymers[22,23], molecular metal complexes[24–26], covalent organic frameworks (COFs)[27–29], and metal-organic frameworks (MOFs)[8,30]. However, most studies have focused on detecting single HCl vapors by relying on a protonation mechanism, which cannot

---

[1]Department of Chemistry Education, Seoul National University, Seoul, Republic of Korea. [2]These authors contributed equally: Wonhyeong Jang, Hyejin Yoo. ✉e-mail: jykim@snu.ac.kr

differentiate between various acid vapors (Supplementary Table 1). Recent studies on colorimetric sensor arrays have demonstrated that integrating the detection results from multiple sensor domains can distinguish various chemical vapors, including some acid vapors; however, this requires additional complex data processing[31,32]. Consequently, despite their importance, research on sensor materials capable of visually identifying acidic gases remains in the preliminary stages.

To develop a true optical molecular decoder for the identification of colorless acidic vapors, an anion-participating colorimetric sensing mechanism is required. Transition-metal chelate complexes are attractive candidates for use in visual identification sensors. Chelating ligands such as ethylenediaminetetraacetic acid (EDTA, $C_{10}H_{16}N_2O_8$), which form a stable metal complex, can release chelated metal ions under acidic conditions[33–35]. The de-chelated transition metal ion acts as a colorimetric center, displaying characteristic colors depending on the coordinating ligands and coordination geometry, thus enabling the visualization of acidic vapors in a single-domain sensor. However, most pure metal-chelate complexes exist as non-porous powders, which impedes their direct exposure to acid vapors and their processing into practical sensors.

Herein, we report a MOF-based acid vapor decoder, MOF-808-EDTA-Cu, capable of visually identifying exposed colorless acid vapors (Fig. 1).

MOF-808, a robust and porous Zr-based MOF with the ideal structural formula $Zr_6O_4(OH)_4(BTC)_2(HCOO)_6$ (BTC denotes 1,3,5-benzenetricarboxylic acid), was selected as the platform, and Cu-chelated EDTA (Cu-EDTA) was incorporated as the proton-triggered colorimetric center (Supplementary Fig. 1). Direct exposure of the Cu-EDTA colorimetric center to HCl vapors facilitated the cyan-to-yellow color transition at HCl concentrations as low as 120 ppm. Interestingly, approximately 80% of the Cu-EDTA colorimetric center could be regenerated through simple immersion in water, allowing the sensor to be reused up to three times while retaining half of its initial Cu content. Furthermore, the Cu-EDTA-grafted MOF-808 exhibited characteristic color changes from cyan to pale green, dark purple, and brown upon exposure to different acid vapors (HF, HBr, and HI), whereas it did not react to interfering gases, humidity, or temperature variations. This acid-selective colorimetric behavior originates from the de-chelation of metal ions from the stable Cu-EDTA, triggered by protons, followed by a color-change mechanism involving anions in the acid vapors. To the best of our knowledge, this approach enables the specific identification of acid vapor by utilizing both protonation and anion-dependent systems within a single domain sensor. Due to the strong chelating properties of EDTA, multiple metal ions could be easily incorporated into a single sensor domain with atomic-level dispersion, expanding the decoding capability to identify six different acidic vapors. Additionally, by introducing a flexible polymer, the Cu-EDTA-decorated MOF-808 was successfully processed into a portable miniaturized acid decoder, exhibiting distinct color changes detectable by the naked eye and monitored by camera sensors. This proves its practicality and versatility as an acid vapor-triggered colorimetric decoder for 24-hour on-site monitoring applications.

## Results

### Preparation and characterization of MOF-808-EDTA-Cu

To develop an acid vapor decoder system featuring a fully exposed colorimetric decoder, EDTA-coordinated MOF-808 (designated as MOF-808-EDTA) was prepared following a previously reported method[36,37] (Supplementary Fig. 2, 3). The colorimetric center $Cu^{2+}$ was introduced by soaking MOF-808-EDTA in a 100 mM $Cu^{2+}$ solution for 24 h, affording a cyan-colored powder. The Cu-ion-incorporated MOF-808-EDTA was denoted as MOF-808-EDTA-Cu (Fig. 2a). The X-ray powder diffraction (XRPD) patterns and SEM image of MOF-808-EDTA-Cu confirmed that the framework structure and morphology was

maintained even after post-synthetic treatments (Fig. 2b, Supplementary Fig. 2–4). Inductively coupled plasma-atomic emission spectrometry (ICP-AES) confirmed that the $Cu^{2+}$ ions capable of chelating up to 82% of EDTA were successfully incorporated into MOF-808-EDTA-Cu (Supplementary Tables 2, 3). Concurrently, the Fourier transform-infrared (FT-IR) spectrum of MOF-808-EDTA-Cu showed the absence of peaks associated with $NO_3^-$ constituting the Cu precursor, ruling out the possibility of the physical inclusion of Cu precursors into the MOF pores (Supplementary Fig. 5)[38,39]. Upon the inclusion of Cu ions into MOF-808-EDTA, the ultraviolet-visible near-infrared (UV-vis-NIR) spectrum revealed a new absorption peak at $13140\ cm^{-1}$, corresponding to the $d$-$d$ transition $^2E_g \rightarrow {}^2T_{2g}$ of an octahedral six-coordinated Cu ion, which is the same as in the previously reported $[Cu(EDTA)(H_2O)]$[40], suggesting the presence of Cu-EDTA in the MOF-808-EDTA-Cu system (Supplementary Fig. 6). X-ray photoelectron spectroscopy (XPS) of MOF-808-EDTA-Cu (Supplementary Fig. 7) revealed a significant shift in the Zr $3d$ binding energy compared to MOF-808, with the monocarboxylate ligand removed, indicating that the EDTA molecule chelating the Cu ion remained grafted to the $Zr_6$ cluster of MOF-808-EDTA-Cu[37,41]. Nitrogen sorption measurements of MOF-808-EDTA-Cu showed a decrease both in surface area and pore size of larger cavity ($1118\ m^2/g$ and 0.81 nm) compared to that of the pristine MOF-808 ($2065\ m^2/g$ and 1.28 nm), indicating that Cu-EDTA, the colorimetric center, exists in accessible internal pores of MOF-808 rather physical mixed (Fig. 2c and Supplementary Fig. 8). Therefore, MOF-808-EDTA-Cu containing fully exposed Cu-EDTA within its accessible pores, was successfully prepared as a colorimetric sensor via a simple post-synthetic modification.

### Colorimetric response of MOF-808-EDTA-Cu to HCl vapor

An interesting naked-eye detectable color change from cyan to yellow was observed in MOF-808-EDTA-Cu within 20 s of exposure to HCl vapors evaporated from a concentrated HCl solution (Fig. 2e and Supplementary Movie 1). This transition occurred without any structural changes in the MOF-808 framework, suggesting that the eye-detectable color shift did not originate from structural decomposition (Fig. 2b). Further analysis confirmed the incorporation of chlorine into MOF-808-EDTA-Cu after exposure, as evidenced by scanning electron microscopy coupled with energy-dispersive X-ray spectroscopy (SEM-EDS) (Fig. 2d) and the XPS spectra of Cl $2p$ (Supplementary Fig. 9). The emergence of a Cu−Cl stretching vibration peak at ~$289\ cm^{-1}$ in the Raman spectra (Supplementary Fig. 10) indicated the formation of new bonds between Cu and Cl[42]. Upon exposure to HCl, UV-vis−NIR spectroscopy showed new absorption peaks at 34364 and $25316\ cm^{-1}$, suggesting that the coordination environment of the $Cu^{2+}$ ion is changed (Fig. 2f). Interestingly, these peaks align with the LMCT characteristics reported for tetrahedrally structured $CuCl_4^{2-}$[2,43,44], implying that free $Cu^{2+}$ ions de-chelated from EDTA forming a new yellow Cu-Cl complex (Fig. 2f). Importantly, MOF-808-EDTA-Cu demonstrated a selective colorimetric response towards acid gas, maintaining its original cyan color unchanged after 24 hours of exposure to potentially interfering air gases such as $N_2$, $O_2$, and $CO_2$, as well as variations in humidity and temperature within a substantial range (Supplementary Fig. 11). This highlights its potential as a reliable acid-vapor-selective sensor for practical applications.

An intriguing colorimetric response was observed upon exposure of MOF-808-EDTA-Cu to HCl vapor, which was detectable even at concentrations as low as 120 ppm, although a longer detection time was required. Cu-EDTA and $Cu(CH_3COO)_2$, characterized by non-porous and densely packed structures, displayed gradual color changes after acidic exposure for 1 h at 120 ppm. Conversely, MOF-808-EDTA-Cu exhibited a visible color shift within 5 min, underscoring the significance of grafting Cu-EDTA onto MOF-808 and directly expose it to external acidic vapors for efficient acid sensing (Supplementary Fig. 12). Upon immersion in water, the yellow color of the HCl-exposed

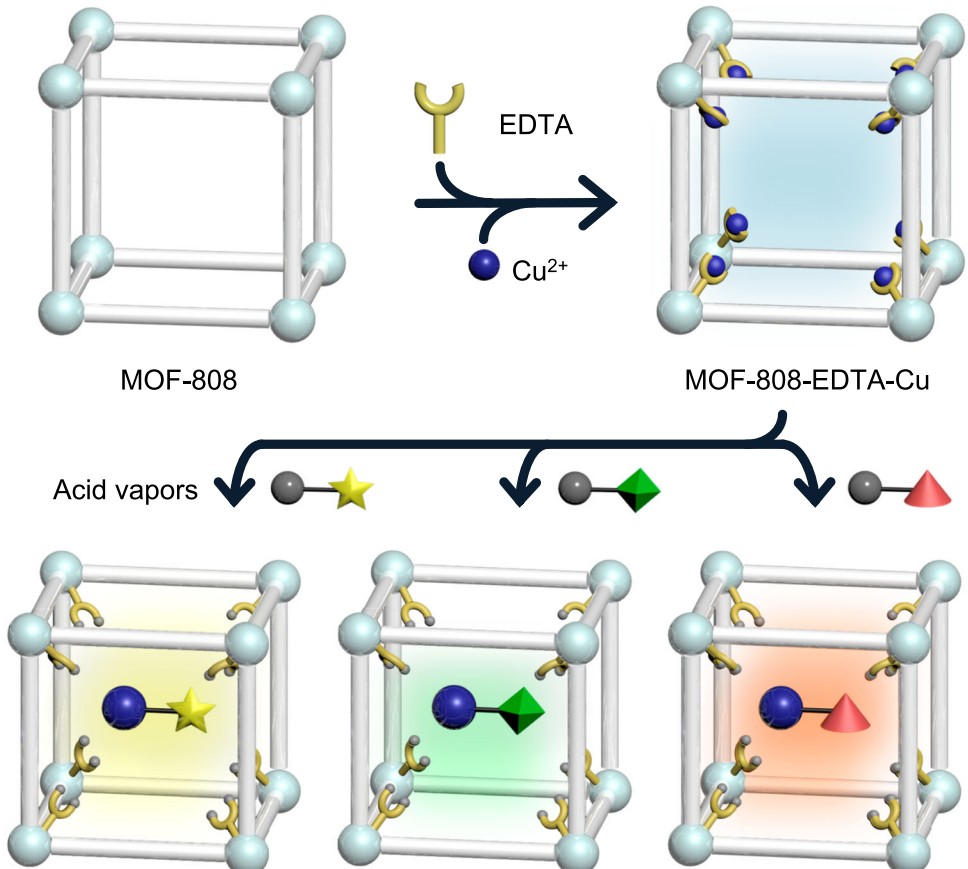

**Fig. 1 | Schematic illustration of the preparation of MOF-808-EDTA-Cu and its application as a colorless acid vapor sensor.** MOF-808-EDTA-Cu was synthesized by employing MOF-808 as the substrate and incorporating EDTA and $Cu^{2+}$ through PSM. It exhibits a distinct visible color change in response to varying acids, integrating both protonation and anion-participation mechanisms.

MOF-808-EDTA-Cu immediately changed to cyan and was recovered as a cyan powder through filtration (Fig. 2e). The recovered MOF exhibited the consistent UV-Vis-NIR spectrum as that of MOF-808-EDTA-Cu (Supplementary Fig. 13). This intriguing regeneration of the sensor is attributed to the characteristics of EDTA, which is known to efficiently chelate various metal ions even at low concentrations in aqueous solutions[37,45,46]. Consequently, MOF-808-EDTA demonstrated an ability to re-chelate approximately 80% of $Cu^{2+}$ ions during the regeneration process, as confirmed by ICP-AES analysis (Supplementary Table 2). Furthermore, during of alternating exposure to HCl and water, the color of MOF-808-EDTA-Cu faded slightly with each cycle but still clearly exhibited reversible cyan-yellow color variations observable to the naked eye, highlighting the reusability of the sensor (Fig. 2e, Supplementary Fig. 13 and Supplementary Table 2).

**Acid-triggered colorimetric decoding mechanism**
To reveal the underlying mechanism of the formation of yellow Cu-Cl complexes from highly stable cyan-colored Cu-EDTA complexes, a series of controlled experiments were conducted with MOF-808-EDTA-Cu. First, to elucidate the formation conditions of the Cu-Cl complexes from MOF-808-Cu-EDTA, three types of aqueous solutions containing the same 4 M $Cl^-$ ions were prepared: two neutral (NaCl and KCl) and one acidic (HCl) solution. In the 4 M $Cl^-$ solution, a portion of the introduced free $Cu^{2+}$ ions formed a yellow Cu-Cl complex, leading to a color change from blue to greenish-yellow (Supplementary Fig. 14a). MOF-808-EDTA-Cu turned greenish-yellow only in the acidic HCl solution, while retaining its cyan color in the other solutions, indicating the generation of free $Cu^{2+}$ ions from Cu-EDTA only under acidic conditions (Supplementary Fig. 14b). This acid-condition-

selective color transformation stems from alterations in the functional groups of EDTA, as evidenced by the FT-IR spectra of MOF-808-EDTA-Cu before and after exposure to HCl (Fig. 2g). Compared to MOF-808-EDTA-Cu, the HCl-exposed MOF-808-EDTA-Cu showed decreased intensity of the peak at 1566 cm$^{-1}$, corresponding to the $v_{as,COO^-}$ of EDTA[47–50], while the emergence of a new peak at 1719 cm$^{-1}$ was attributed to the $v_{C=O}$ of carboxylic acids, indicating the protonation of the carboxylate in EDTA[34,51]. XPS spectra of the HCl-exposed MOF-808-EDTA-Cu compared to MOF-808-EDTA-Cu revealed new peaks at 401.6 and 533.1 eV corresponding to the N 1$s$ of the protonated amine ($R_3NH^+$) and O 1$s$ of the carboxylic acid in EDTA[52–54], respectively, further supporting the protonation of EDTA which hardly chelates the $Cu^{2+}$ ion, resulting in the release of free Cu ions (Fig. 2h and Supplementary Fig. 15). Therefore, both $Cl^-$ and $H^+$ are essential for the colorimetric decoding of acid vapors by MOF-808-EDTA-Cu. This acid-selective color transition mechanism demonstrates the potential of MOF-808-EDTA-Cu as a true colorimetric acid sensor that can selectively react with anions in acidic environments.

**MOF decoder to visualize exposed acid vapors**
Building on the acid-triggered and anionic participation in the colorimetric sensing mechanism, we explored the potential of MOF-808-EDTA-Cu as a colorimetric sensor for the visualization of colorless hydrohalic acid vapors. The MOF-808-EDTA-Cu sensor distinctly visualized the exposed hydrohalic acid vapors, namely HF, HBr, and HI, as white, dark purple, and brown, respectively (Fig. 3a). Interestingly, these observed color changes were aligned with the expected colors resulting from the interaction of free Cu ions with halide ions, suggesting that the de-chelated $Cu^{2+}$ ions from the protonated EDTA in

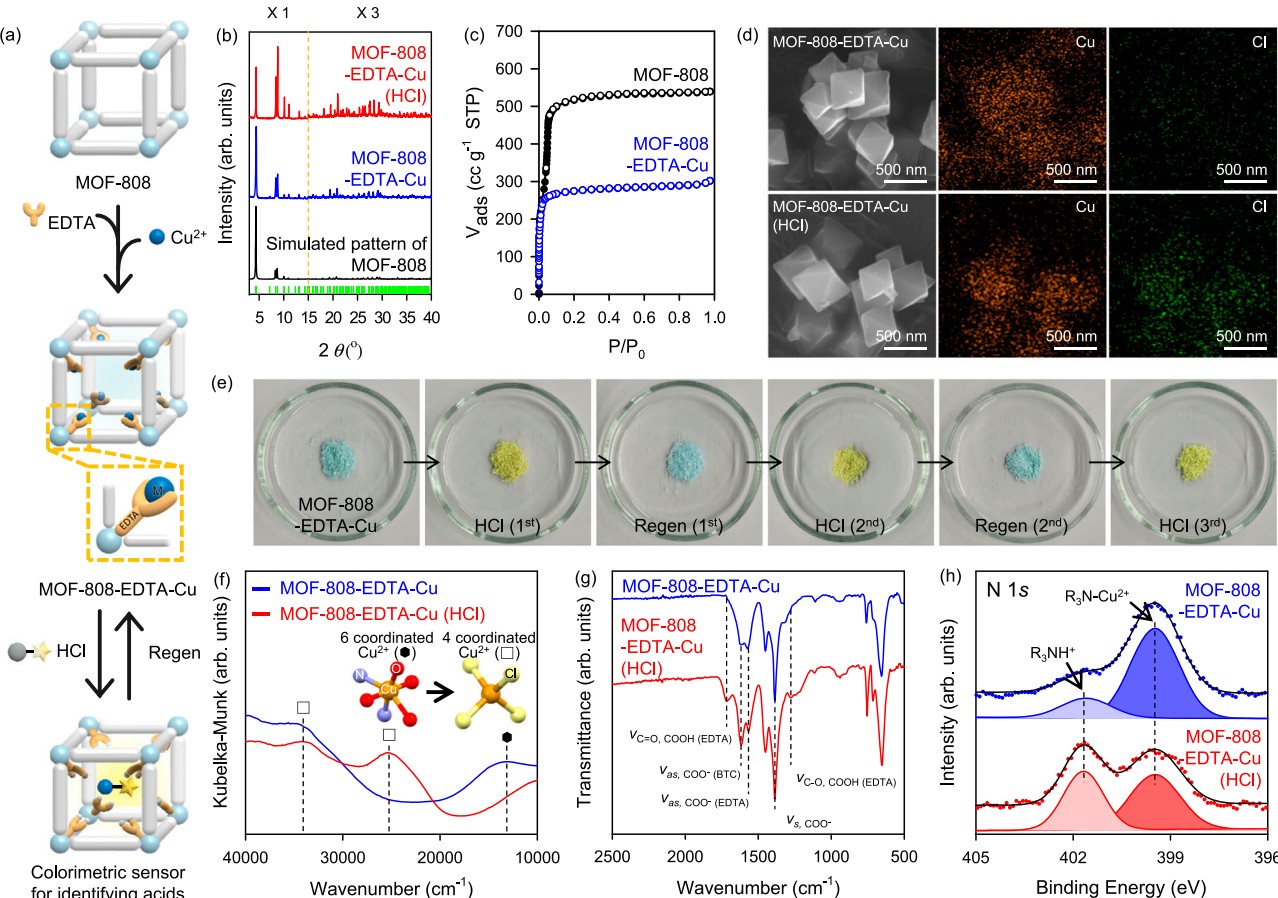

**Fig. 2 | Preparation and Characterization of MOF-808-EDTA-Cu with Colorimetric Response to HCl Vapor. a** Schematic illustration of the preparation of MOF-808-EDTA-Cu as an acid vapor decoder and its visual identification of hydrochloric acid vapor. **b** XRPD patterns of MOF-808-EDTA-Cu (blue) and MOF-808-EDTA-Cu (HCl) (red) with simulated XRPD pattern of MOF-808 (black). The intensity was tripled for the range of 2θ from 15 to 40 degrees, with an orange dash line marking 15 degrees. **c** N₂ sorption isotherms of MOF-808 (black) and MOF-808-EDTA-Cu (blue) measured at 77 K. Filled and open symbols correspond to

adsorption and desorption, respectively. **d** SEM images and the EDS elemental mapping images of MOF-808-EDTA-Cu (top) and MOF-808-EDTA-Cu (HCl) (bottom). **e** Photographs of MOF-808-EDTA-Cu with hydrochloric acid vapor exposure and regeneration series. **f** Diffuse reflectance UV-vis-NIR spectra of MOF-808-EDTA-Cu (blue) and MOF-808-EDTA-Cu (HCl) (red). Inset: Suggested coordination system of Cu²⁺, copper (orange), oxygen (red), nitrogen (violet), and chlorine (yellow). **g** FT-IR spectra of MOF-808-EDTA-Cu (blue) and MOF-808-EDTA-Cu-HCl (red). **h** N 1s XPS spectra of MOF-808 EDTA-Cu (top) and MOF-808 EDTA-Cu (HCl) (bottom).

MOF-808-EDTA-Cu reacted with hydrohalic acids (Supplementary Fig. 16). Specifically, the UV-vis-NIR spectra of MOF-808-EDTA-Cu after exposure to HF and HBr were consistent with those of CuF₂ and CuBr₄²⁻, respectively[55], implying the formation of white CuF₂ and purple CuBr₄²⁻ within the MOF sensor following acid exposure (Supplementary Fig. 17, 18). Furthermore, exposure to HI resulted in the formation of white CuI(s) and brown I₃⁻, as confirmed by PXRD and UV-vis-NIR spectra, suggesting that the brown color of the HI-exposed MOF-808-EDTA-Cu originated from I₃⁻ rather than CuI(s) (Supplementary Fig. 19). In addition, immersion of the acid-exposed MOF-808-EDTA-Cu in water restored its cyan color, thereby enabling recovery as a cyan powder through filtration (Supplementary Fig. 20). The exceptionally strong chelation of EDTA in MOF-808-EDTA renders it versatile and enables the incorporation of various metal ions into the MOF-808 sensor. To further explore our methodology, we prepared Fe³⁺-chelated MOF-808-EDTA (referred to as MOF-808-EDTA-Fe) as a colorimetric sensor, which exhibited a distinct color change from ivory to yellow and orange upon exposure to HCl and HBr, respectively, confirming the formation of FeCl₄⁻ and FeBr₄⁻ within the sensor after acid exposure[56] (Fig. 3b and Supplementary Fig. 21). The UV-vis-NIR spectra of MOF-808-EDTA-Fe after exposure to HCl and HBr were consistent with those of FeCl₄⁻ and FeBr₄⁻, respectively[56], implying the

formation of FeCl₄⁻ and FeBr₄⁻ within the MOF sensor following acid exposure (Supplementary Fig. 21).

Furthermore, the strong chelation capability of EDTA allowed MOF-808-EDTA to effectively integrate multiple metal ions (Cu²⁺ and Co²⁺) with atomic-level dispersion, thereby expanding the scope of visually-identifiable acid vapors within a single-domain sensor (Fig. 3c). MOF-808-EDTA-Cu/Co, featuring Cu-chelated EDTA and Co-chelated EDTA, exhibited an ability to differentiate between six acidic vapors within a single-domain sensor (Fig. 3d). This capability arises from the co-presence of Cu-chelated EDTA, adept at decoding hydrohalic acid and Co-chelated EDTA, proficient at decoding nitric acid and trifluoroacetic acid (TFA). Such findings demonstrate the versatility and resilience of our sensor platform, providing visual identification of a variety of acid vapors with a single sensor domain.

To exploit the properties of the MOF-808-EDTA-Cu platform for the visualization of colorless acid vapors, the fabrication of miniaturized portable acid vapor sensors that could be used for real-time on-site monitoring was explored (Fig. 4a). For transformation into a portable acid vapor decoding sensor, a MOF sensor-based ink was fabricated by combining MOF-808-EDTA-Cu with dimethylformamide (DMF) solution containing polyvinylidene fluoride (PVDF),

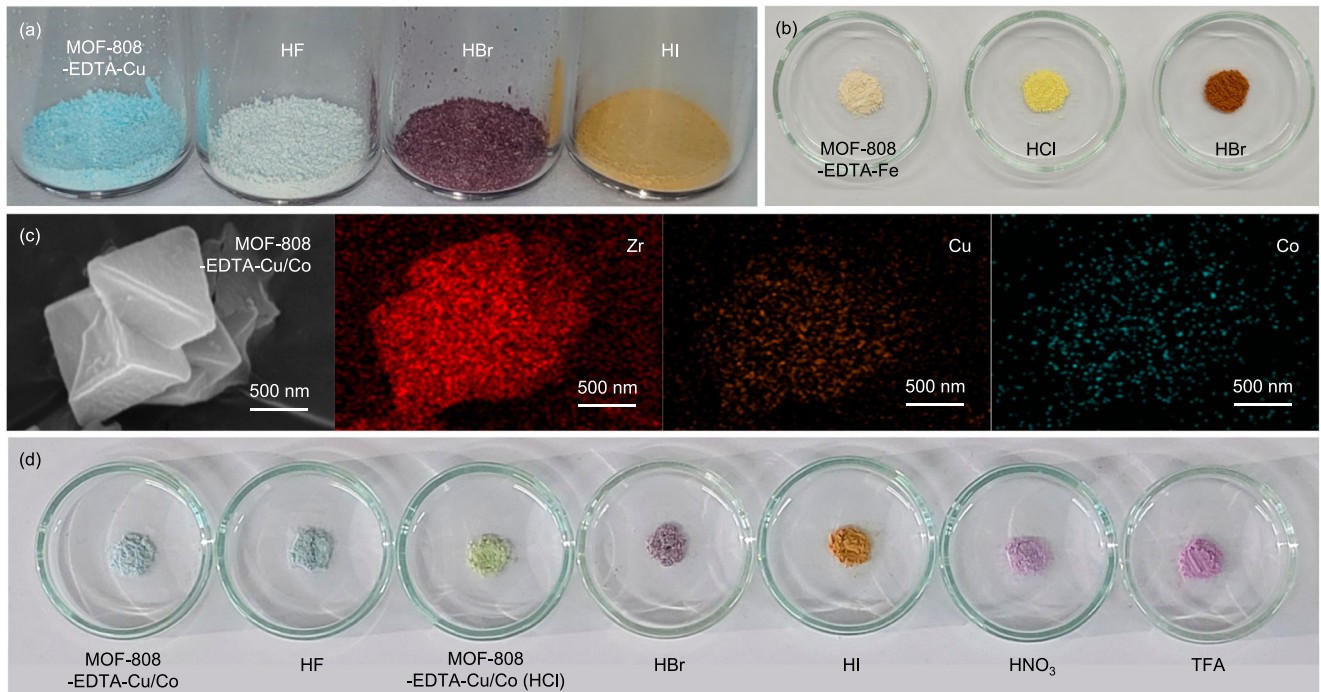

**Fig. 3 | MOF-808-EDTA-Metal for acid vapor identification. a** Photographs of MOF-808-EDTA-Cu for identifying HCl, HBr, and HI vapors. **b** Photographs of MOF-808-EDTA-Fe after exposure to HCl and HBr vapors. **c** SEM images and the EDS elemental mapping images of MOF-808-EDTA-Cu/Co (Cu atomic% = 2.91%, Co atomic% = 1.34%). **d** Photographs of MOF-808-EDTA-Cu/Co after exposure to HF, HCl, HBr, HI, HNO$_3$, and trifluoroacetic acid vapors.

allowing high MOF loading and ensuring the surface remains accessible to the environment[57,58]. The MOF sensor-based ink with optimized PVDF amounts of 20 wt% was applied to various substrates, including foil, paper, fabric, and glass (Supplementary Figs. 22, 23). When exposed to HCl vapor evaporating from a concentrated HCl solution (approximately 15,500 ppm)[59,60], the MOF-808-EDTA-Cu portable sensor underwent a distinct color shift from cyan to yellow. This cyan-yellow color variation persisted over three cycles of alternating exposure to HCl and water (Supplementary Fig. 24). The distinct cyan-to-yellow color change was detectable by the camera sensor and translated into RGB channel values (Fig. 4b), allowing the quantification of the color changes and 24-hour real-time monitoring. This color change was further translated into RGB channel values, allowing the quantification of the color changes and 24-hour real-time monitoring. Notably, when exposed to low concentrations of HCl where the color change is not saturated, the sensor exhibited a reduced transition from cyan to yellow within the same exposure timeframe, suggesting its potential as an acid–gas concentration analyzer (see Fig. 4c). This transition can be precisely quantified using the equation $|dB|/B_0$, where $|dB|$ denotes the absolute value of change in the blue channel value from the initial blue channel value $B_0$. Correlation of the $|dB|/B_0$ ratio with different HCl vapor concentrations can establish a linear range spanning from 120 to 740 ppm, providing experimental validation of the portable sensor as a colorimetric sensor capable of quantifying the concentration of exposed HCl vapor. Additionally, when exposed to an atmosphere with high relative humidity (RH) of 85%, no color change was detected both with the naked eye or even with RGB values, demonstrating its practical use as a portable sensor capable of selectively visualizing acidic vapors, even in the presence of humidity interference (Fig. 4d).

Based on the obtained results, we extended our investigation on the color changes of sensors upon exposure to various acid vapors. In experiments with hydrohalic acid vapors, including HF, HBr, and HI, the color change of the MOF-808-EDTA-Cu portable sensor with 20 wt%

PVDF was not complete until 25 min, however, changes detectable by the naked eye appeared within 10 min (Supplementary Fig. 25). Interestingly, when monitoring the color shifts of the portable sensor via the RGB channel values, distinct trends in the RGB channel values depending on the exposed acid vapors were observed, even in the early stages when they were barely detectable by the naked eye (Fig. 4e). Furthermore, these acid-dependent distinct color alterations enable the statistical validation of exposure to hydrohalic acid vapor within 2 min by applying principal component analysis (PCA) and hierarchical cluster analysis (HCA) methods. As shown in Fig. 4f, the 12 datasets of dR, dG, and dB obtained from three repeated 2-min exposure experiments with four different hydrohalic acids formed distinct clusters that were well-spaced apart, implying efficient identification. HCA-based data classification using Ward's method revealed that when the closest data points were clustered, three points originating from the same acid vapor exposure experiments were successfully grouped together, confirming the ability of the sensor to discriminate between acids (Fig. 4g). Moreover, this versatile portable sensor platform can incorporate various transition metals, such as Co and Fe, to broaden the decoding range of acid vapors to up to six types or to adjust the detection color (Supplementary Fig. 26), providing experimental validation of its applicability to diverse industrial requirements.

In conclusion, we successfully fabricated a colorimetric acid vapor sensor capable of on-site differentiation between various acid vapors, leveraging the color-changing attributes of a built-in colorimetric center, Cu-EDTA, in the robust and porous MOF-808-EDTA-Cu. Cu-EDTA grafted in MOF-808-EDTA-Cu was directly exposed to acid vapors, enhancing its effectiveness in identifying acid vapors and translating the different anion components of corrosive acids into visible colors. Notably, MOF-808-EDTA-Cu was unresponsive to interfering gases, humidity, and temperature variations, rendering it a practical and versatile acid-triggered sensor for on-site applications. This distinctive acid-selective colorimetric behavior stems from the proton-triggered de-chelation of metal ions from the stable Cu-EDTA, which is followed by a color-change mechanism that involves anions

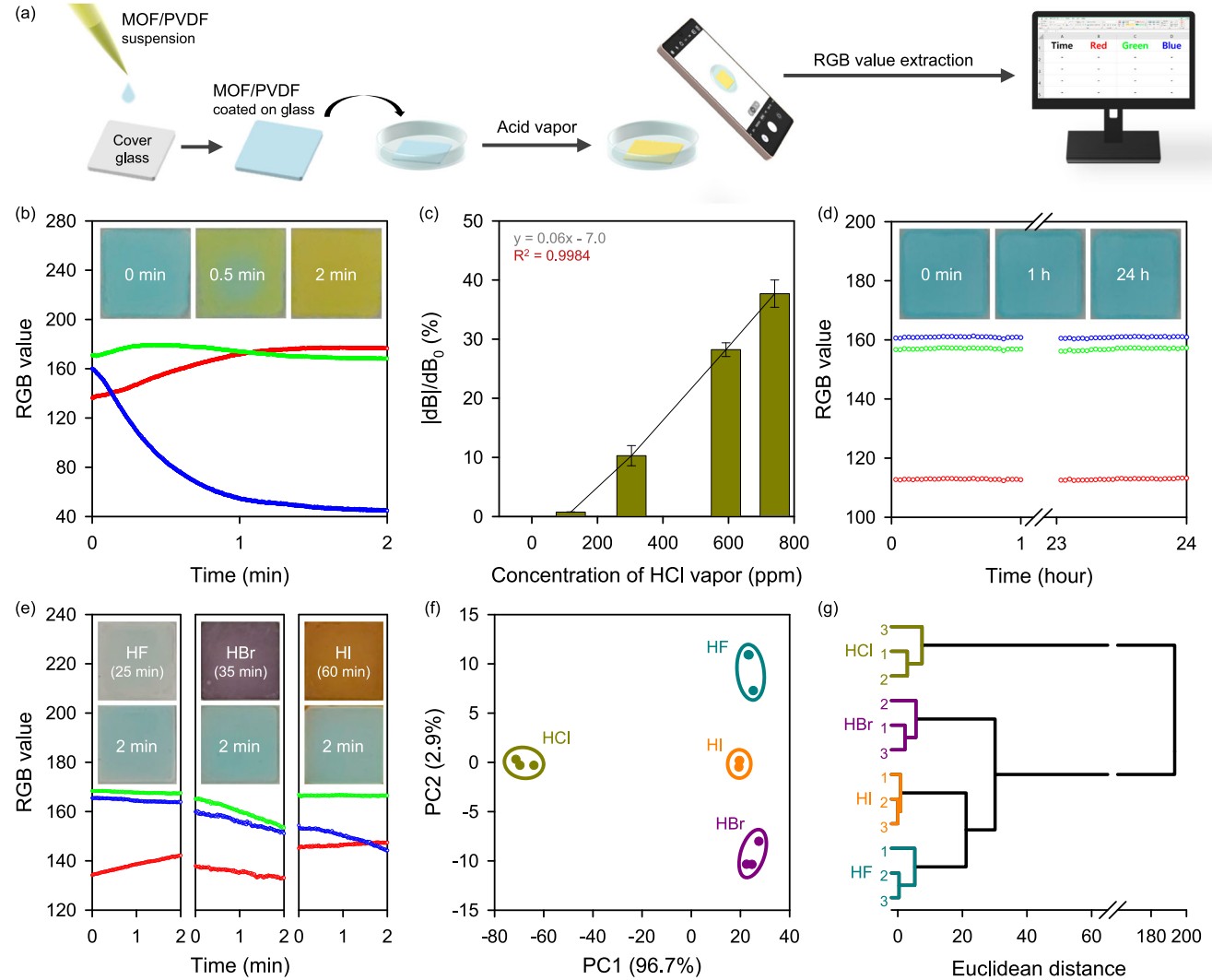

**Fig. 4 | MOF-808-EDTA-Cu portable sensor for on-site acid vapor monitoring.**
**a** Schematic illustration of the preparation of MOF-808-EDTA-Cu film and process for the extraction of RGB (Red, Green, Blue) channel value of the film. **b** Time-dependent RGB curves of MOF-808-EDTA-Cu portable sensor with 20 wt% PVDF exposed to 15500 ppm HCl vapor. Inset: Photographs of MOF-808-EDTA-Cu portable sensor with 20 wt% PVDF under 15500 ppm HCl vapor exposure. **c** Response curve by the relationship between $|dB|/dB_0$ value and concentration of HCl vapor (120, 300, and 740 ppm) after 3 min of acid exposure. The error bar states the standard error based on three experimental trials. **d** Time-dependent RGB curves of MOF-808-EDTA-Cu portable sensor with 20 wt% PVDF exposed to 85% RH water vapor for 24 hours. Inset: Photographs of MOF-808-EDTA-Cu portable sensor with 20 wt% PVDF under 85% RH water vapor exposure. **e** Time-dependent RGB curves of MOF-808-EDTA-Cu portable sensors with 20 wt% PVDF exposed to HF, HBr, and HI vapor. Inset: Photographs of the MOF-808-EDTA-Cu portable sensor with 20 wt% PVDF after exposure to HF, HBr, and HI vapors for 25 min, 35 min, and 60 min, respectively (top), and after 2 min of exposure (bottom). **f** Principal component analysis (PCA) results for discrimination of hydrohalic acids, based on three experimental trials. **g** Hierarchical cluster analysis (HCA) dendrogram designed for categorizing hydrohalic acids, based on three experimental trials.

present in the acid vapors. This mechanism, which involves both the proton and anion of the acid, allows not only the detection of the presence of acid but also its visual distinction, which is a distinct advantage over traditional acid vapor sensors[8,18–30]. Further, the strong chelating properties of EDTA in MOF-based sensors enable the easy extension to other metal ions, such as Fe and Co, broadening its ability to detect various acid vapors in a single-domain sensor as well as its potential customization for specific industrial needs. The integration of a polymer into the MOF sensor led to the development of a portable miniaturized sensor capable of visually identifying six different colorless acid vapors, highlighting its versatility in the practical 24-hour on-site monitoring of acid vapor sensor applications. This simple yet sophisticated decoding method using wireless communication technology enables the development of gas sensors capable of detection and identification of hazardous acid chemicals, providing comprehensive and real-time data for large-scale environmental monitoring.

## Methods

### Materials and Characterization

1,3,5-Benzentricarboxylic acid (95%), sulfuric acid-$d_2$ (96–98%), hydrofluoric acid (48%), copper(II) nitrate trihydrate (99–104%), iron(III) nitrate nonahydrate (98%), trifluoroacetic acid (99%), poly(vinylidene fluoride) (average MW: ~ 534,000), copper(II) acetate monohydrate (98%), sodium chloride (99%), potassium chloride (99%), copper(I) iodide (98%) were purchased from Sigma-Aldrich. Zirconium dichloride oxide octahydrate (98%), formic acid (99%), cobalt(II) nitrate hexahydrate (98–102%), hydrobromic acid (48%), hydroiodic acid (55–58%) were purchased from Thermo Fisher Scientific. Dimethyl sulfoxide-$d_6$ (99.9%) was purchased from Cambridge Isotope Laboratories. Nitric acid (60%), N,N-dimethylformamide (99.5%), acetone (99.5%), ethylenediaminetetraacetic acid disodium salt (99%), hydrochloric acid (35–37%) were purchased from DAEJUNG chemicals. Copper(II) fluoride (98%) was purchased from Tokyo Chemical

Industry. All chemicals and solvents were of reagent grade and used without further purification.

X-ray powder diffraction (XRPD) patterns were collected on a Bruker D2 PHASER at 30 kV and 10 mA for Cu K$_\alpha$ ($\lambda$ = 1.54050 Å), with a step size of 0.02° in 2$\theta$. Fourier transform-Infrared (FT-IR) spectra were recorded on a Bruker ALPHA II FT-IR spectrometer using the attenuated total reflection (ATR) mode. $^1$H nuclear magnetic resonance (NMR) spectra were measured on a Bruker Advance III HD 300 MHz. For NMR sample preparation, 0.005 g of samples were digested using D$_2$SO$_4$ (20 μL) and DMSO-d$_6$ (600 μL) as solvents. The nitrogen adsorption-desorption isotherm was obtained using a Quantachrome Instruments Autosorb-iQ at 77 K. All samples (~ 60 mg) were activated under ultra-high vacuum at 130 °C for 24 h prior to each measurement. The surface areas were calculated using BETSI, following the Rouquerol criteria 1–4[61]. Pore size distribution was calculated using quenched solid density functional theory (QSDFT) method. Scanning electron microscopy (SEM) images and energy dispersive X-ray spectroscopy (EDS) mapping were taken using JSM 7800 F Prime operating at 15 kV. For SEM imaging, the samples were placed on the carbon tape on an aluminum sample holder and coated using carbon-sputter coating. X-ray photoelectron spectroscopy (XPS) data were obtained by using an AXIS SUPRA and spectra were analyzed using XPSPEAK 4.1. Inductively coupled-atomic emission spectroscopy (ICP-AES) data were collected on a Perkin Elmer Optima 8300. For ICP-AES sample preparation, 0.01 g of samples were digested with 60 μL of hydrofluoric acid. The hydrofluoric acid was completely removed by vaporization before the samples were further dissolved with 4 mL of nitric acid. The acid-digested samples were diluted with deionized water before measurement. UV-Vis-NIR spectra were recorded with a PerkinElmer Lambda 365 UV/Vis spectrophotometer for reflectance measurement. Raman spectroscopy data were obtained using a Thermo Fischer Scientific DXR2xi Raman imaging microscope with 532 nm laser source.

## Synthesis of MOF-808-EDTA

MOF-808-EDTA was prepared based on the methods reported on previous literature studies[36,37]. 1,3,5-Benzentricarboxylic acid (0.786 g, 3.7 mmol) and ZrOCl$_2$·8H$_2$O (1.209 g, 3.7 mmol) were dissolved in the mixture of N,N-Dimethylformamide (DMF) (150 mL) and formic acid (150 mL) in a 500 mL lab bottle, and the bottle was heated in an oven at 130 °C for 24 h. The white powder was collected by filtration and washed with DMF twice daily for three days. It was then soaked in water for three days, with the water being replaced twice a day. Finally, the process was repeated with acetone. The MOF-808 was activated by heating at 150 °C for 24 h in vacuum condition. Yield: 0.834 g (67%), $^1$H-NMR (D$_2$SO$_4$/DMSO-d$_6$): δ 8.56 (s, 3H), FT-IR (ATR, cm$^{-1}$): $v_{\text{as (carboxylate, BTC)}}$ = 1605(s), $v_{\text{as (carboxylate, formate)}}$ = 1564(s), $v_{\text{O-C=O(aromatic carboxylate, sym)}}$ = 1445(s), $v_{\text{O-C=O(aliphatic carboxylate, sym)}}$ = 1379(s). Then 0.100 g of activated MOF-808 and 1.860 g of ethylenediaminetetraacetic acid disodium salt (EDTA-2Na) were dissolved in 50 mL water. The contents were placed in a 100 mL lab bottle and heated at 80 °C for 24 h. The powder was filtered and washed with water several times to remove unreacted EDTA. It was then washed several times with fresh acetone. The solid was dried overnight at 100 °C under vacuum. $^1$H-NMR (D$_2$SO$_4$/DMSO-d$_6$): δ 8.56 (s, 3H), δ 4.10 (s, 8H), δ 3.60 (s, 4H), FT-IR (ATR, cm$^{-1}$): $v_{\text{O-C=O(aromatic carboxylate, asym)}}$ = 1615(s), $v_{\text{O-C=O(aliphatic carboxylate, asym)}}$ = 1566(s), $v_{\text{O-C=O(aromatic carboxylate, sym)}}$ = 1445(s), $v_{\text{O-C=O(aliphatic carboxylate, sym)}}$ = 1381(s), $v_{\text{C-N(EDTA)}}$ = 1214(sh).

## Synthesis of MOF-808-EDTA-Metal

To prepare MOF-808-EDTA-M (M = Cu or Fe), MOF-808-EDTA (0.100 g) was added into a glass vial containing 10 mL aqueous solution of 0.1 M metal nitrate (Cu(NO$_3$)$_2$·3H$_2$O and Fe(NO$_3$)$_3$·9H$_2$O for MOF-808-EDTA-Cu and MOF-808-EDTA-Fe, respectively). The mixture was stirred at room temperature for 24 h and filtered through a 0.2 μm

polytetrafluoroethylene (PTFE) membrane filter. The resulting powder was activated by heating at 100 °C for 24 h in a vacuum condition and stored at ambient conditions.

To prepare MOF-808-EDTA-Cu/Co, MOF-808-EDTA (0.100 g) was added into 10 mL of mixed metal solution containing equal concentration (50 mM) of Cu(NO$_3$)$_2$·3H$_2$O and Co(NO$_3$)$_2$·6H$_2$O. The mixture was stirred at room temperature for 24 h and filtered through a 0.2 μm PTFE membrane filter. The resulting powder was activated by heating at 100 °C for 24 h in the vacuum condition and stored at ambient conditions.

## Acid vapor detection and regeneration of MOF-808-EDTA-metal

A 3 cm glass dish containing 0.010 g of MOF-808-EDTA-Metal was placed inside a 5 cm glass dish with 2 mL of concentrated HCl solution, ensuring no direct contact between the MOF-808-EDTA-Metal and the acid solution. The 5 cm dish was covered with a 7 cm glass dish to detect vaporized acid. For HF, HBr, HI, HNO$_3$, and CF$_3$COOH vapor detection experiments, 2 mL of respective concentrated acid solutions were used instead of concentrated HCl solution. Notably, the HF solution was handled using a polystyrene petri dish to avoid contact with the glass.

For different concentration of HCl vapor detection, varying concentrations of HCl vapor were prepared using HCl solutions with different wt% based on a previously reported method[59,60]; 15460 ppm, 740 ppm, 590 ppm, 300 ppm and 120 ppm HCl vapor were prepared using 37.1 wt%, 24.7 wt%, 22.0 wt% 21.6 wt% and 18.5 wt% HCl solutions, respectively.

To regenerate the acid vapor-exposed MOF-808-EDTA-Metal, the powder was added to 10 mL of DI water and stirred for 5 minutes. It was then filtered through a 0.2 μm membrane filter. The resulting powder was dried overnight at 100 °C under vacuum.

## Preparation of portable and miniaturized MOF-808-EDTA-metal sensor

The portable and miniaturized MOF-808-EDTA-Metal sensor was fabricated by combining MOF-808-EDTA-Metal with polyvinylidene fluoride (PVDF) based on the previous literature with minor modifications[62]. 0.048 g of MOF-808-EDTA-Metal was dispersed in 2.4 mL of acetone and sonicated for 30 minutes in a vial. Then, 0.6 mL of a DMF solution containing 0.012 g of PVDF (M$_w$-534,000), yielding a PVDF concentration of 20 wt%, was added to the MOF suspension. The suspension was further sonicated for 30 minutes, and the acetone was removed using rotary evaporation at 20 °C under a 100 mbar vacuum for 10 minutes, yielding MOF sensor-based ink. A 200 μL aliquot of this ink was drop-cast onto various substrates, including foil, paper, fabric, and glass, and then dried in an 80 °C oven for 1 hour. The portable sensors with 10 wt% and 40 wt% PVDF were fabricated in the same way as above, utilizing 0.005 g and 0.032 g of PVDF, respectively.

## On-site monitoring of portable and miniaturized MOF-808-EDTA-metal sensor combined with smartphone camera

The MOF-808-EDTA-Metal portable sensor was prepared by coating 200 μL of MOF sensor-based ink containing 20 wt% PVDF onto a 1.8 cm x 1.8 cm cover glass. This prepared MOF portable sensor was exposed to acid vapors under the same conditions as the acid vapor detection for MOF-808-EDTA-Metal, replacing the MOF-808-EDTA-Metal powder with the MOF portable sensor. During the acid vapor exposure, the color change of the MOF portable sensor was recorded as a video using the standard camera application on the Galaxy S20. The camera's autofocus and exposure settings were locked during recording. The video was taken in a commercially available photobox with a white LED light (6000 K color temperature and 95 color rendering index (CRI)) to ensure consistent conditions and minimize the influence of surrounding light. To quantify the color changes of the portable sensor recorded in the video, a Python script utilizing the OpenCV library was

run in Linux to crop the video featuring the MOF portable sensor and calculate the mean RGB channel values of the cropped frames as a function of exposure time. To regenerate the MOF-808-EDTA-Metal portable sensor, it was completely submerged in a glass dish containing 10 mL of water, removed, and allowed to vacuum dry.

## Discrimination analysis

Principal component analysis (PCA) and hierarchical cluster analysis (HCA) were performed by using R 4.3.3 programming language, operating with R studio. A matrix with a size of $12 \times 3$ (three trials of four different acids x dR, dG, dB value) was entered as input data without relying on sample labels. For PCA, 'prcomp' function was mainly used to decrease the dimensionality of data, forming new principal components. The new principal component space was plotted to show as a score plot. For HCA, 'hclust' function was mainly used to facilitate data categorization. The dendrogram was generated for the distance method as Euclidean and the cluster method as Ward's method.

## Data availability

The data that support the findings of this study are presented in the main manuscript and the Supplementary Information. Additional data is available on request from the corresponding authors. Source data are provided with this paper.

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

## Acknowledgements

Research in the J.Y.K. laboratory is supported by a National Research Foundation (NRF) of Korea funded by the Ministry of Science and ICT, South Korea (NRF-2022R1C1C101022013), POSCO Science Fellowship of POSCO TJ Park Foundation, and 'Science Education in Infosphere(SEI)' of the four-stage BK21.

## Author contributions

W.J. and J.Y.K. conceived the idea and designed the experiments. W.J. and H.Y. synthesized the molecular structures, carried out the experimental work, and analyzed the data; D.S. helped the synthesis of molecular structures experiments. S.N. helped in the N₂ sorption isotherms and PXRD measurements. W.J., H.Y. and J.Y.K. co-wrote the manuscript. All authors discussed and analyzed the results.

## Competing interests

The authors declare no competing interests.
