## [Transparent Peer Review file · Nature Communications]

Colorimetric Identification of Colorless Acid Vapors using a Metal-Organic Framework-Based Sensor

Corresponding Author: Professor Jin Yeong Kim

Version 0:

Reviewer comments:

Reviewer #1

(Remarks to the Author)

The manuscript reports on a colorimetric acid vapor sensor utilizing MOF-808-EDTA-Cu as the sensing material. The study explores the sensing mechanism, which is based on the proton-triggered de-chelation of metal ions from EDTA. Although this work provides an effective approach for the detection of acid vapor, the present manuscript is not suitable for acceptance due to several issues that need to be addressed. I recommend that this contribution be considered for publication after the following issues are resolved. The detailed comments are listed below.

1. To emphasize the significance of the designed sensor utilizing MOF-808-EDTA-Cu, it is crucial to compare it with previously reported acid vapor-sensing materials. Please include these comparisons.
2. ICP-AES is mentioned several times in the manuscript, but specific results are not provided. Why? Please include this data in detail.
3. In Fig. 1e, MOF-808-EDTA-Cu demonstrated reversibility over three cycles. Why specifically three cycles or more?
4. How reversible is the MOF/PVDF film? It is highly recommended to investigate this aspect.
5. The manuscript lacks a clear description of the MOF ink coating method on the substrate. Please clarify this process.
6. It is not clear why PVDF was chosen as the polymer matrix and whether the mass of PVDF affects the sensing performance of the MOF/PVDF sensor. Please provide further explanations.
7. While Fig. 3d tests the humidity stability of MOF/PVDF, the test duration is too short, only 5 minutes. How is the long-term stability under varying humidity conditions?
8. In the part of "Methods-Synthesis of MOF-808-EDTA", it is not three days respectively but three times. Please check and correct it.
9. It is intriguing that the response time is only 20 seconds. It is highly recommended to provide a video recording of the reaction process.

Reviewer #2

(Remarks to the Author)

In this manuscript, the authors propose a novel on-site monitorable acid vapor decoder, MOF-808-EDTA-Cu, integrating the robust MOF-808 with Cu-EDTA. This system functions as a proton-triggered colorimetric decoder that translates the anionic components of corrosive acids into visible colors. The sensor exhibits a remarkable color change when exposed to HX vapor and can visually differentiate various acidic vapors (HF, HBr, and HI) through unique color changes. Furthermore, the compatibility of the MOF-based sensor with multiple metal ions having atomic-level dispersion broadens its discrimination range, enabling the identification of six different colorless acid vapors within a single sensor domain. By incorporating a flexible polymer, the MOF-808-EDTA-Cu has been successfully processed into a portable miniaturized acid sensor, exhibiting distinct color changes that can be easily monitored by the naked eye and camera sensors. This provides experimental validation as a practical sensor capable of on-site 24-hour monitoring in real-world conditions. These fascinating results will attract significant attention from researchers in the fields of both materials science and chemistry and deserve publication in this journal after minor revision.

The reviewer's concerns focus on the points below:

- (1) In Figure S5, the binding energy of Zr 3d_{5/2} and 3d_{3/2} in the XPS spectra of MOF-808-EDTA-Cu shifts to lower energy compared to MOF-808, and no signals of MOF-808 are detected. This indicates that all Zr₆ cluster nodes were coordinated by EDTA. However, the analysis of N₂ adsorption isothermal plots demonstrated that only a small amount of EDTA was incorporated into the channels of MOF-808. What causes this divergence?

(2) How does relative humidity in the air influence the sensitivity of MOF-808-EDTA-Cu to hydrogen halides?

Reviewer #3

(Remarks to the Author)

In this manuscript, the authors reported the synthesis of new materials MOF-808-EDTA-M and its colorimetric sensing and identification ability toward acid vapors. New materials MOF-808-EDTA-Cu, -Fe, and -Cu/Co were synthesized by post-synthetic modification of a known MOF-808-EDTA. The structure of MOF-808-EDTA-Cu was identified to be Cu(η^5 -edta)(H₂O) complex bound to the Zr₆ cluster of MOF-808 via Zr-carboxylate coordination. The vapor acidochromism of MOF-808-EDTA-Cu by HCl was demonstrated, its reversibility and selectivity were evaluated, and its kinetics were compared with solid Cu(edta)(H₂O) and Cu(OAc)₂. The mechanism of acidochromism was elucidated to be the formation of [CuCl₄]²⁻ complex assisted by the protonation of edta. The acid-dependency of the acidochromism of MOF-808-EDTA-Cu, -Fe, and -Cu/Co was demonstrated for HF, HCl, HBr, HI, HNO₃, and TFA. A sensor device was fabricated using MOF-808-EDTA-M with an optical camera, and its ability to sense and distinguish acid vapors aided by hierarchical cluster analysis.

The results provide significant insights to the fields of metal-organic frameworks and vapor sensing. The concept of colorimetric identification of acid vapors and the mechanism which allows it in this system are novel. The major conclusions are well-supported by experimental evidence, and there are no major flaws in the discussion. The characterization methods are mostly sound, but a few more data would help improve the understanding of the system, and the description of methods should be supplemented to facilitate replication of the results. This work can be recommended for publication in Nature Communications, given the following points are addressed.

Line 36: Here are the previous reports on colorimetric sensors for acid vapors are listed, but they seem to lack coordination compounds besides MOFs, such as molecular metal complexes. Since coordination chemistry has central importance in this manuscript, the references to the related literature should be augmented. For example, some of these papers from the Xiang-Jun Zheng Group can be cited:

Liang, Q.-F., Zheng, H.-W., Yang, D.-D., Zheng, X.-J. Zn(II) complexes based on a schiff base: Mechanochromism- and solvent molecule-dependent acidochromism. *Cryst. Growth Des.* 22, 3924-3931 (2022).

Liang, Q.-F., Zheng, H.-W., Yang, D.-D., Zheng, X.-J. A triphenylamine derivative and its Cd(II) complex with high-contrast mechanochromic luminescence and vapochromism. *CrystEngComm* 24, 543-551 (2022).

Yang, D.-D. et al. Multi-stimuli responsive behavior of two Schiff base complexes with high contrast multicolor switching and wearable applications for rapid detection of HCl and NH₃ vapor. *Dyes and Pigments* 212, 111149 (2023).

It would be also worth noting such color-changing phenomena behind sensing are sometimes denoted as acidochromism and/or vapochromism.

Line 80: Although the syntheses of MOF-808 and -EDTA are previously reported, it is not always a trivial task to reproduce MOF synthesis. Therefore, the product should be characterized in more detail. For this purpose, the PXRD patterns in Figure 1b and Supplementary Figure 1 require magnification of the high-angle region to allow inspection of this area. The N₂ sorption isotherm should be compared with a previous report or a theoretical structural model. The origin of the formic acid signal in the NMR spectrum should be identified. The amount of EDTA confirmed by NMR or other techniques should be stated, which is important for sensing properties. The particle morphology should be characterized by SEM or other methods, not only to check amorphous impurities but also because it can affect the sensing kinetics and processability.

Lines 85 and 145: Here the amount of Cu²⁺ ions in MOF-808-EDTA-Cu is stated, which is important for sensing properties. However, the information on how this value is calculated based on ICP-AES measurement is missing. How is the coordination per EDTA calculated from the Cu weight fraction obtained by ICP-AES? Is the possibility of Cu²⁺ binding at different sites such as defects or SBUs considered?

Line 98: There is no information on how the surface areas are calculated. Are they calculated with the BET theory with robust criteria (ref. <http://doi.org/10.1002/adma.202201502>)?

Line 104, panel h: The "NH⁺" and "C-N" signals could be more accurately labeled as "R₃NH⁺" and "R₃N-Cu²⁺".

Line 131: How stable is the color change after switching the environment to the air?

Line 135: Here is the importance of the porous MOF structure demonstrated by comparison with pure Cu(edta)(H₂O) solids. However, if only the porosity is important, it is possible the MOF-808 support can be replaced by other cheaper porous materials. For example, cobalt(II) chloride impregnated in silica gel or paper is known as a colorimetric sensor of water vapor. Does MOF-808 have advantages over these traditional porous materials?

Line 143: Here the efficiency of regenerating the sensor material is discussed. Does this value depend on the amount of water used? Can this value be improved by using a buffer, base, or NH₃ vapor for effectively removing HCl?

Line 145: Here it is stated this sensor material shows reversible color changes. However, the experiment is done only for three cycles, and the UV-vis spectra show some irreversible changes. It would be better to state this partial reversibility quantitatively.

Line 176: Are the changes after exposure to HF, HBr, or HI gas also (partially) reversible?

Line 183: Here the formation of "I₂(aq)" is discussed, although there is no aqueous medium in this experiment. Is this a mistake of I₂ adsorbed in the material? How is the product identified as I₂? Does the color fade over time by the evaporation of I₂?

Line 189: Is there any information about the acidochromism mechanism of MOF-808-EDTA-Fe? For example, do the UV-vis absorption peaks match the LMCT transitions of [FeX₄]⁻?

Line 197: Are the cobalt ions in this material confirmed to be Co²⁺ but not Co³⁺?

Line 201: Is there any information about the chromism mechanism of MOF-808-EDTA-Cu/Co by nitric acid and TFA? Was the used nitric acid pure and free from NO₂?

Line 209: Is the fabricated sensor device reusable as shown in the case of the powder form? Is the color retained after exposure to air?

Line 280: There is no information on the used materials and chemicals such as purity and supplier, which are important for replication of this work.

Line 285: The procedures for digesting solid samples for solution NMR spectroscopy are missing.

Line 307 and others: The ^1H NMR chemical shifts are reported as values in "DMSO-d6", which is likely to be a mistake of D₂SO₄/DMSO-d₆ with a certain ratio.

Line 328: The procedures for acid vapor detection are described only for HCl, and those for other acids are missing.

SI Line 33: There is no information on how the pore size distributions are calculated.

SI Line 74 panel b: The PXRD pattern after HF exposure shows a higher background and noise. Does this indicate the decomposition of the MOF-808 framework by HF, which is reasonable for Zr-MOFs?

General: It is recommended to attach adsorption data in the adsorption information file (AIF) format as supplementary materials or upload them to a data repository for accurate inspection and usage by readers.

Version 1:

Reviewer comments:

Reviewer #1

(Remarks to the Author)

In the revised version, most issues have been addressed. However, further revisions are highly recommended to enhance the manuscript before it is published in Nature Communications. The details are as follows.

Q1: To emphasize the significance of the designed sensor utilizing MOF-808-EDTA-Cu, it is crucial to compare it with previously reported acid vapor-sensing materials. Please include these comparisons.

Although the authors added four new references in revised manuscript, the significance of the designed sensor utilizing MOF-808-EDTA-Cu it still not emphasized. It is strongly recommended to include a comparison table to better highlight the sensing performance of the designed sensor.

Q3: In Fig.1e, MOF-808-EDTA-Cu demonstrated reversibility over three cycles. Why specifically three cycles or more?

The findings are intriguing, as they reveal a decrease in copper content during the reversibility process. It is recommended to include this evidence in the revised manuscript. It would provide valuable insight and offer a useful approach for exploring the reversibility of sensing materials.

Q6: It is not clear why PVDF was chosen as the polymer matrix and whether the mass of PVDF affects the sensing performance of the MOF/PVDF sensor. Please provide further explanations.

Although the PVDF content was optimized, the specific optimal amount is not clearly indicated in the revised manuscript. Please include this information.

Reviewer #2

(Remarks to the Author)

I am satisfied with the authors' revisions and responses to the comments provided. I recommend accepting this manuscript for publication in Nature Communications in its current form.

Reviewer #3

(Remarks to the Author)

The authors revised the manuscript thoroughly, and they addressed most of the points I and the other reviewers raised. This work is recommended for publication, given the following points are addressed.

1. My previous question, "10. Are the changes after exposure to HF, HBr, or HI gas also (partially) reversible?" was answered properly, but the manuscript was not revised accordingly. Since it is an advantage of this material that it can reversibly detect these gases, I believe it is worth noting.

2. In response to my question 20, the authors described that the pore size distribution had been calculated using the Horvath–Kawazoe (HK) method. However, this method is obsolete and specific to graphitic slit pores in microporous carbons, which do not resemble the pores of MOF-808. This choice of method can be the reason that the pore size distribution in Figure S7 does not match the crystal structure or that in reference S5. The pore calculation should be conducted instead with an NLDFT or QSDFT method, which is more universal, accurate, and used in reference S5. Since the isotherms are recorded with an Autosorb sorptometer, the software ASiQwin can be used for such analysis.

3. I noticed that the figures in the main text contain no chemical structure. The structures of important chemicals such as MOF-808, EDTA, and [Cu(edta)(OH₂)] would better be depicted to help the readers understand the system.

Version 2:

Reviewer comments:

Reviewer #1

(Remarks to the Author)

There are no further comments.

Reviewer #3

(Remarks to the Author)

The authors properly addressed the reviewers' comments. I recommend the current manuscript for publication without any further revision.

Responses to the Comments of Reviewer #1

(Reviewer's Comments) *The manuscript reports on a colorimetric acid vapor sensor utilizing MOF-808-EDTA-Cu as the sensing material. The study explores the sensing mechanism, which is based on the proton-triggered de-chelation of metal ions from EDTA. Although this work provides an effective approach for the detection of acid vapor, the present manuscript is not suitable for acceptance due to several issues that need to be addressed. I recommend that this contribution be considered for publication after the following issues are resolved.*

(Author's Response) Thank you for taking the time to provide us with such a constructive review and valuable feedback on our manuscript. We have carefully addressed the issues raised by the reviewer in the attached responses. We hope the reviewer finds these satisfactory and thank you again for the reviewer's time and effort. Below, we have written a point-by-point response to reviewer's comments.

1. **(Reviewer's Comments)** *To emphasize the significance of the designed sensor utilizing MOF-808-EDTA-Cu, it is crucial to compare it with previously reported acid vapor-sensing materials. Please include these comparisons.*

(Author's Response) We thank the reviewer's valuable comments on comparisons of previously reported acid vapor-sensing materials. As suggested, we have revised the sentence and added more examples to be more clarified and make the comparisons illuminating.

Changes made:

- We have modified the sentence and added four new references in revised manuscript as below.

→(Line 37) "Recently, several colorimetric sensors for the detection of acid vapors have been reported utilizing organic dyes¹⁸⁻¹⁹, polymers²⁰, molecular metal complexes²¹⁻²³, covalent organic frameworks (COFs)²⁴⁻²⁶, and metal-organic frameworks (MOFs)^{8,27}."

New references:

21. Liang, Q.-F., Zheng, H.-W., Yang D.-D. & Zheng, X.-J. Zn(II) complexes based on a Schiff base: mechanochromism- and solvent molecule-dependent acidochromism. *Cryst. Growth Des.* **22**, 3924-3931 (2022).
22. Yang, D.-D. et al., Multi-stimuli responsive behavior of two Schiff base complexes with high contrast multicolor switching and wearable applications for rapid detection of HCl and NH₃ vapor. *Dyes and Pigments* **212**, 111149 (2023).
23. Liang, Q.-F., Zheng, H.-W., Yang, D.-D. & Zheng, X.-J. A triphenylamine derivative and its Cd(II) complex with high-contrast mechanochromic luminescence and vapochromism. *CrystEngComm* **24**, 543-551 (2022).
24. Kundu, P. K., Olsen, G. L., Kiss, V. & Klajn, R. Nanoporous frameworks exhibiting multiple stimuli responsiveness. *Nat. Commun.* **5**, 3588 (2014)

- We have added a new sentence to provide further distinction between our research from previously reported acid vapor-sensing materials in revised manuscript.

→ (Line 277) "This mechanism, which involves both the proton and anion of the acid, allows not only the detection of the presence of acid but also its visual distinction, which is a distinct advantage over traditional acid vapor sensors^{8,18-27}."

2. *ICP-AES is mentioned several times in the manuscript, but specific results are not provided. Why? Please include this data in detail.*

Thanks for the valuable suggestions for improving the amiability of the paper. Following the reviewer’s recommendation, we have revised the manuscript and provided the ICP-AES data in the revised supplementary information.

Changes made:

• We have revised sentence in revised manuscript and newly added supplementary table 1 and 2 in revised supplementary information.

→ (Line 86) “Inductively coupled plasma-atomic emission spectrometry (ICP-AES) ~ incorporated into MOF-808-EDTA-Cu (Supplementary Table 1 and 2).”

→ (Line 146) “Consequently, the EDTA-decorated MOF-808 ~ as confirmed by ICP-AES analysis (Supplementary Table 1).”

Supplementary Table 1. ICP-AES analysis results for MOF-808-Cu, MOF-808-EDTA-Cu, and MOF-808-EDTA-Cu-regen.

	Zr		Cu		Zr : Cu (molar ratio)
	mg/L	mmol/L	mg/L	mmol/L	
MOF-808-EDTA-Cu	78.457	0.860	13.193	0.208	1.000 : 0.242
MOF-808-EDTA-Cu-Regen (1 st)	17.093	0.187	2.265	0.036	1.000 : 0.193
MOF-808-EDTA-Cu-Regen (2 nd)	21.414	0.235	2.185	0.034	1.000 : 0.145
MOF-808-Cu ^a	84.745	0.929	0.079	0.001	1.000 : 0.001

^a MOF-808 (without EDTA) was immersed in 10 mL of a 0.1 M aqueous Cu²⁺ solution for 24 hours. ICP-AES analysis showed a nearly negligible amount of Cu²⁺ in MOF-808-Cu, indicating that the majority of Cu²⁺ present in MOF-808-EDTA-Cu is due to the chelation effect of EDTA.

Supplementary Table 2. Calculation of the percentage of Cu²⁺ chelated by EDTA in MOF-808-EDTA-Cu.

	EDTA : BTC ^{a,b}	BTC ^a : Zr ^c	Zr : Cu ^d	EDTA : Cu
MOF-808-EDTA-Cu	0.89 : 1	1 : 3	3 : 0.72	0.89 : 0.72 (82%)

^a Abbreviation for 1,3,5-benzenetricarboxylic acid

^b The ratio of EDTA to BTC confirmed by NMR results (Supplementary Fig. X)

^c The ratio of BTC to Zr in MOF-808 (Zr₆O₄(OH)₄(BTC)₂(HCOO)₆)

^d The ratio of Zr to Cu determined by ICP-AES results (Supplementary Table 1)

3. *In Fig. 1e, MOF-808-EDTA-Cu demonstrated reversibility over three cycles. Why specifically three cycles or more?*

We thank the reviewer for bringing this issue. In Fig. 1e, we demonstrated the reversibility of MOF-808-EDTA-Cu over three cycles to ensure the accuracy and reliability of the reported data. While MOF-808-EDTA-Cu shows reusability over five cycles, as indicated by the PXRD patterns and UV-vis spectra (Reviewer only Figure 1), continuous color fading was observed

due to the 80% re-chelation of Cu^{2+} ions, as mentioned in the original paper. Additionally, ICP-AES analysis revealed a decrease of approximately 50% in copper content after four cycles (Reviewer-only Table 1). Based on these findings, we focused on three cycles to highlight the sensor's reliable performance under the tested conditions.

Reviewer only Figure 1. (a) Photographs of MOF-808-EDTA-Cu during a series of hydrochloric acid vapor exposure and regeneration cycles. (b) XRPD patterns of MOF-808-EDTA-Cu at each stage throughout five cycles of alternating exposure to HCl vapor and water. (c) Diffuse reflectance UV-vis-NIR spectra of MOF-808-EDTA-Cu at various stages: initial, after the 1st HCl exposure, after the 1st regeneration, after the 2nd HCl exposure, after the 4th regeneration, and after the 5th HCl exposure.

Reviewer only Table 1. ICP-AES analysis results comparing the copper content in MOF-808-EDTA-Cu before and after each regeneration cycle.

	Zr		Cu		Zr : Cu (molar ratio)
	mg/L	mmol/L	mg/L	mmol/L	
MOF-808-EDTA-Cu	78.457	0.860	13.193	0.208	1.000 : 0.242
MOF-808-EDTA-Cu-Regen (1 st)	17.093	0.187	2.265	0.036	1.000 : 0.193
MOF-808-EDTA-Cu-Regen (2 nd)	21.414	0.235	2.185	0.034	1.000 : 0.145
MOF-808-EDTA-Cu-Regen (3 rd)	21.163	0.232	1.873	0.029	1.000 : 0.125
MOF-808-EDTA-Cu-Regen (4 th)	13.761	0.151	0.650	0.010	1.000 : 0.066

4. *How reversible is the MOF/PVDF film? It is highly recommended to investigate this aspect.*

We thank the reviewer for bringing this issue to our attention. The MOF/PVDF film exhibits partial reversibility, supporting multiple reuses. To investigate this further, we have conducted additional experiments and updated our manuscript and supplementary information accordingly.

Changes made:

- We have revised sentences in revised manuscript and newly added supplementary figure 22 in revised supplementary information.

→ (Line 218) “When exposed to HCl vapor evaporating from a concentrated HCl solution (approximately 15,500 ppm)⁵⁶⁻⁵⁷, the MOF-808-EDTA-Cu portable sensor underwent a distinct color shift from cyan to yellow. This cyan-yellow color variation persisted over three cycles of alternating exposure to HCl vapor and water (Supplementary Fig. 22). The distinct cyan-to-yellow color change was detectable by the camera sensor and translated into RGB channel values (Fig. 3b), allowing the quantification of the color changes and 24-hour real-time monitoring.”

→ (Line 390) “To regenerate the MOF-808-EDTA-Metal portable sensor, it was completely submerged in a glass dish containing 10 mL of water, removed, and allowed to vacuum dry.”

Supplementary Figure 22. Photographs of MOF-808-EDTA-Cu portable sensor during a series of hydrochloric acid vapor exposure and regeneration cycles.

- The manuscript lacks a clear description of the MOF ink coating method on the substrate. Please clarify this process.

Thank you for the reviewer’s recommendation. Following the suggestion, the sentences were revised to clearly describe the MOF ink coating process as follows:

Changes made:

- We have revised the sentences as follow.

→ (Line 372) “The suspension was further sonicated for 30 minutes, and the acetone was then removed using rotary evaporation at 20 °C under a 100 mbar vacuum for 10 minutes, yielding MOF sensor-based ink. A 200 μ L aliquot of this ink was drop-cast onto various substrates, including foil, paper, fabric, and glass, and then dried in an 80 °C oven for 1 hour.”

- It is not clear why PVDF was chosen as the polymer matrix and whether the mass of PVDF affects the sensing performance of the MOF/PVDF sensor. Please provide further explanations.

Thank you for the reviewer’s insightful question regarding the choice of PVDF as the polymer matrix and its potential impact on the sensing performance. The successful fabrication of a portable acid decoding sensor requires a polymer matrix that can support high MOF loading while ensuring the MOF surface remains accessible to the environment. Polyvinylidene fluoride (PVDF) was selected for this purpose due to its ability to incorporate a high weight percentage of MOF in various MOF-PVDF composites, while preserving the MOF surface’s accessibility (Denny Jr, M. S. et al., *Angew. Chem. Int. Ed.* **54**, 9029-9032 (2015); DeCoste, J. B. et al. *Chem. Sci.* **7**, 2711-2716 (2016).)

The wt% of PVDF significantly influences the sensor’s performance (Supplementary Figure 21). A low wt% of PVDF resulted in poor film formation, while a high wt% of PVDF led to a diminished cyan-to-yellow transition within the same exposure timeframe. Based on these

findings, we optimized the PVDF content to 20wt% in the MOF-polymer composites. The manuscript has been revised accordingly.

Changes made:

- We have revised the sentence as follows;

→ (Line 212) “For transformation into a portable acid vapor decoding sensor, a MOF sensor-based ink was fabricated by combining MOF-808-EDTA-Cu with dimethylformamide (DMF) solution containing polyvinylidene fluoride (PVDF), allowing high MOF loading and ensuring the surface remains accessible to the environment⁵⁴⁻⁵⁵. The MOF sensor-based ink with optimized PVDF amounts was applied to various substrates, including foil, paper, fabric, and glass (Supplementary Figs. 20 and 21).”

New references:

54. Denny Jr, M. S. & Cohen, S. M. In situ modification of metal–organic frameworks in mixed-matrix membranes. *Angew. Chem. Int. Ed.* **54**, 9029-9032 (2015).

55. DeCoste, J. B., Denny Jr, M. S., Peterson, G. W., Mahle, J. J. & Cohen, S. M. Enhanced aging properties of HKUST-1 in hydrophobic mixed-matrix membranes for ammonia adsorption. *Chem. Sci.* **7**, 2711-2716 (2016).

- We have added a recipe for a portable acid vapor decoding sensor with different amounts of PVDF as follows;

→ (Line 375) “0.048 g of MOF-808-EDTA-Metal was dispersed in 2.4 mL of acetone and sonicated for 30 minutes in a vial. Then, 0.6 mL of a DMF solution containing 0.012 g of PVDF (Mw ~ 534,000) was added to the MOF suspension. ~ The portable sensors with 10wt% and 40wt% PVDF were fabricated in the same way as above, utilizing 0.005 g and 0.032 g of PVDF, respectively.”

- We have newly added the Supplementary Figure 21 as follow:

Supplementary Figure 21. Photographs of portable acid vapor decoding sensors processed on cover glass with varying PVDF amounts and after exposure to 15,500 ppm HCl vapor. A low PVDF mass ratio of 10wt% resulted in poor film formation, while a high PVDF mass ratio of 40wt% led to a diminished cyan-to-yellow transition within the same exposure timeframe.

7. While Fig. 3d tests the humidity stability of MOF/PVDF, the test duration is too short, only 5

minutes. How is the long-term stability under varying humidity conditions?

We appreciate the reviewer's concern regarding the humidity stability test duration. To address this, we conducted an extended stability test for 24 hours under the same experimental conditions. The RGB channel values remained stable over 24 hours at 85% RH, indicating that the MOF/PVDF sensor maintains its performance under prolonged humidity exposure. Consequently, we have revised Fig. 3d to reflect this long-term stability.

Changes made:

- We have revised the Fig. 3d in the revised manuscript, extending the time from 5 minutes to 24 hours.

Fig. 3 (d) Time-dependent RGB curves of MOF-808-EDTA-Cu portable sensor exposed to 85% RH water vapor for 24 hours. Inset: Photographs of MOF-808-EDTA-Cu portable sensor under 85% RH water vapor exposure.

8. In the part of “Methods-Synthesis of MOF-808-EDTA”, it is not three days respectively but three times. Please check and correct it.

We appreciate the reviewer's comment. To clarify the washing procedure, we have revised the sentence as follows.

Changes made:

- We have revised the sentences as follows;
 - (Line 328) “The white powder was collected by filtration and washed with DMF, water and acetone for three days respectively, during which time the solvents were replaced two times per day.”
→ “The white powder was collected by filtration and washed with DMF twice daily for three days. It was then soaked in water for three days, with the water replaced twice daily. Finally, the process was repeated using acetone.”
9. It is intriguing that the response time is only 20 seconds. It is highly recommended to provide a video recording of the reaction process.

Thank you for the reviewer's interest in the response time. In response to the reviewer's comments, we have included a video recording of the reaction process as Supplementary Movie 1. For the reviewer's convenience, screenshots from Supplementary Movie 1 has been provided in Reviewer only Figure 2 as below:

Changes made:

- We have newly added the Supplementary Movie 1, updated its legend in the description of supporting files and revised the sentence in revised manuscript:

→ (Line 119) “An interesting naked-eye detectable color ~ from a concentrated HCl solution (Fig. 1e and Supplementary Movie 1).”

Supplementary Movie 1: HCl vapor detection of MOF-808-EDTA-Cu. A 3 cm glass dish containing 0.010 g of MOF-808-EDTA-Cu was placed inside a 5 cm glass dish with 2 mL of concentrated HCl solution, ensuring no direct contact between the MOF-808-EDTA-Cu and the acid solution. The 5 cm dish was covered with a 7 cm glass dish to detect vaporized acid. The numbers at the bottom of the video indicate the reaction time.

Reviewer only Figure 2. Screenshots from Supplementary Movie 1 showing the color change of MOF-808-EDTA-Cu upon exposure to HCl vapors generated from a concentrated HCl solution. The numbers at the bottom of the video indicate the reaction duration.

Responses to the Comments of Reviewer #2

(Reviewer's Comments) *In this manuscript, the authors propose a novel on-site monitorable acid vapor decoder, MOF-808-EDTA-Cu, integrating the robust MOF-808 with Cu-EDTA. This system functions as a proton-triggered colorimetric decoder that translates the anionic components of corrosive acids into visible colors. The sensor exhibits a remarkable color change when exposed to HX vapor and can visually differentiate various acidic vapors (HF, HBr, and HI) through unique color changes. Furthermore, the compatibility of the MOF-based sensor with multiple metal ions having atomic-level dispersion broadens its discrimination range, enabling the identification of six different colorless acid vapors within a single sensor domain. By incorporating a flexible polymer, the MOF-808-EDTA-Cu has been successfully processed into a portable miniaturized acid sensor, exhibiting distinct color changes that can be easily monitored by the naked eye and camera sensors. This provides experimental validation as a practical sensor capable of on-site 24-hour monitoring in real-world conditions. These fascinating results will attract significant attention from researchers in the fields of both materials science and chemistry and deserve publication in this journal after minor revision.*

(Authors' Response) We are grateful to the reviewer for the careful review and positive feedback on our study. We have carefully considered the points raised by the reviewer and have attempted to respond to each of them in detail. We trust that our responses are satisfactory and extend our thanks to the reviewer for taking the time and effort to evaluate our paper. Below, we have written a point-by-point response to reviewer's comments.

1. *In Figure S5, the binding energy of Zr 3d_{5/2} and 3d_{3/2} in the XPS spectra of MOF-808-EDTA-Cu shifts to lower energy compared to MOF-808, and no signals of MOF-808 are detected. This indicates that all Zr₆ cluster nodes were coordinated by EDTA. However, the analysis of N₂ adsorption isothermal plots demonstrated that only a small amount of EDTA was incorporated into the channels of MOF-808. What causes this divergence?*

We sincerely thank the reviewer for their constructive comments regarding the potential inconsistency between the XPS and N₂ adsorption data. This apparent divergence arises from the different analytical techniques used. XPS is a surface-sensitive method, probing only the outer ~10 nm of the material, while N₂ sorption measures the internal porosity of the entire MOF-808 crystal.

While the XPS data shows a shift in the binding energy of Zr 3d_{5/2} and 3d_{3/2}, indicating coordination between EDTA and the Zr clusters, it does not reflect the overall amount or distribution of EDTA throughout the MOF-808 crystal. Therefore, as noted in the original manuscript, we confirmed that 1.6 EDTA molecules per Zr cluster were incorporated and homogeneously distributed within the MOF-808 crystal using ¹H-NMR and SEM-EDS mapping analysis. This is consistent with the N₂ adsorption data, which shows that although the N₂ adsorption amounts of MOF-808 decreases after the introduction of Cu-EDTA, the pores remain externally accessible.

2. *How does relative humidity in the air influence the sensitivity of MOF-808-EDTA-Cu to hydrogen halides?*

We are grateful to the reviewer for raising the issue of sensor performance under varying relative humidity. To investigate this, we conducted HCl exposure experiments at three different relative humidity (RH) levels: approximately 95%, 65%, and 35%. As shown in

Reviewer only Figure 3, no significant differences in the performance of the MOF-808-EDTA-Cu sensor were observed across these RH levels. However, the humidity sensor was highly unstable in the presence of HCl gas, preventing accurate RH measurements during the experiments, despite possible fluctuations from the introduction of aqueous HCl. Consequently, we concluded that while our sensor can detect HCl gas in humid environments, the current setup does not allow for reliable detection results corresponding to humidity levels due to the inability to obtain accurate humidity values. For this reason, these results were not included in the main manuscript. Conducting acid exposure experiments under controlled humidity, using advanced equipment such as acid-stable humidity sensors and dedicated acid and humidity controllers, will be a future practical topic for further study, and we hope that the reviewer will agree with this conclusion.

Reviewer only Figure 3. Photographs of MOF-808-EDTA-Cu with hydrochloric acid vapor exposure with three different relative humidity (RH). Relative humidity levels of approximately 95% and 35% were created using wet tissues and silica gel, respectively.

Responses to the Comments of Reviewer #3

(Reviewer's Comments) *In this manuscript, the authors reported the synthesis of new materials MOF-808-EDTA-M and its colorimetric sensing and identification ability toward acid vapors. New materials MOF-808-EDTA-Cu, -Fe, and -Cu/Co were synthesized by post-synthetic modification of a known MOF-808-EDTA. The structure of MOF-808-EDTA-Cu was identified to be Cu(η^5 -edta)(H₂O) complex bound to the Zr₆ cluster of MOF-808 via Zr-carboxylate coordination. The vapor acidochromism of MOF-808-EDTA-Cu by HCl was demonstrated, its reversibility and selectivity were evaluated, and its kinetics were compared with solid Cu(edta)(H₂O) and Cu(OAc)₂. The mechanism of acidochromism was elucidated to be the formation of [CuCl₄]²⁻ complex assisted by the protonation of edta. The acid-dependency of the acidochromism of MOF-808-EDTA-Cu, -Fe, and -Cu/Co was demonstrated for HF, HCl, HBr, HI, HNO₃, and TFA. A sensor device was fabricated using MOF-808-EDTA-M with an optical camera, and its ability to sense and distinguish acid vapors aided by hierarchical cluster analysis.*

The results provide significant insights to the fields of metal–organic frameworks and vapor sensing. The concept of colorimetric identification of acid vapors and the mechanism which allows it in this system are novel. The major conclusions are well-supported by experimental evidence, and there are no major flaws in the discussion. The characterization methods are mostly sound, but a few more data would help improve the understanding of the system, and the description of methods should be supplemented to facilitate replication of the results. This work can be recommended for publication in Nature Communications, given the following points are addressed.

(Authors' Response) We would like to extend our sincerest gratitude to the reviewer kindly taking the time and effort to conduct an evaluation of our paper. We also appreciate the reviewer's in-depth and constructive comments provided. We have made every effort to address all the feedback thoroughly and hope that our responses satisfactorily address the reviewer's concerns. Below, we have written a point-by-point response to reviewer's comments.

1. *Line 36: Here are the previous reports on colorimetric sensors for acid vapors are listed, but they seem to lack coordination compounds besides MOFs, such as molecular metal complexes. Since coordination chemistry has central importance in this manuscript, The references to the related literature should be augmented. For example, some of these papers from the Xiang-Jun Zheng Group can be cited:*

Liang, Q.-F., Zheng, H.-W., Yang, D.-D., Zheng, X.-J. Zn(II) complexes based on a schiff base: Mechanochromism- and solvent molecule-dependent acidochromism. Cryst. Growth Des. 22, 3924-3931 (2022).

Liang, Q.-F., Zheng, H.-W., Yang, D.-D., Zheng, X.-J. A triphenylamine derivative and its Cd(ii) complex with high-contrast mechanochromic luminescence and vapochromism. CrystEngComm 24, 543-551 (2022).

Yang, D.-D. et al. Multi-stimuli responsive behavior of two Schiff base complexes with high contrast multicolor switching and wearable applications for rapid detection of HCl and NH₃ vapor. Dyes and Pigments 212, 111149 (2023).

It would be also worth noting such color-changing phenomena behind sensing are sometimes denoted as acidochromism and/or vapochromism.

We appreciate the reviewer's valuable suggestion. We acknowledge the importance of including references to coordination compounds in the context of colorimetric sensors, particularly in relation to our work on MOFs. Following your recommendations, we have

included the references in the manuscript. Additionally, we have incorporated the terms “acidochromism” and “vapochromism” into the manuscript to better describe the color-changing phenomena observed in our sensors.

Changes made:

- We have cited the suggested references.

→ (Line 37) “Recently, several colorimetric sensors for the detection of acid vapors have been reported utilizing organic dyes¹⁸⁻¹⁹, polymers²⁰, molecular metal complexes²¹⁻²³, covalent organic frameworks (COFs)²⁴⁻²⁶, and metal-organic frameworks (MOFs)^{8,27}.”

New references:

21. Liang, Q.-F., Zheng, H.-W., Yang D.-D. & Zheng, X.-J. Zn(II) complexes based on a Schiff base: mechanochromism- and solvent molecule-dependent acidochromism. *Cryst. Growth Des.* **22**, 3924-3931 (2022).

22. Yang, D.-D. et al., Multi-stimuli responsive behavior of two Schiff base complexes with high contrast multicolor switching and wearable applications for rapid detection of HCl and NH₃ vapor. *Dyes and Pigments* **212**, 111149 (2023).

23. Liang, Q.-F., Zheng, H.-W., Yang, D.-D. & Zheng, X.-J. A triphenylamine derivative and its Cd(II) complex with high-contrast mechanochromic luminescence and vapochromism. *CrystEngComm* **24**, 543-551 (2022).

- We have added new terms, acidochromism and vapochromism, to better describe the color-changing phenomena in the revised manuscript.

→ (Line 34) “One of the most attractive approaches in this field involves the construction of colorimetric molecular decoders that exhibit acidochromism and vapochromism, offering simplicity in identifying acid vapors with the naked eye, cost efficiency, and the potential for on-site identification of multiple targets.”

2. *Line 80: Although the syntheses of MOF-808 and -EDTA are previously reported, it is not always a trivial task to reproduce MOF synthesis. Therefore, the product should be characterized in more detail. For this purpose, the PXRD patterns in Figure 1b and Supplementary Figure 1 require magnification of the high-angle region to allow inspection of this area. The N₂ sorption isotherm should be compared with a previous report or a theoretical structural model. The origin of the formic acid signal in the NMR spectrum should be identified. The amount of EDTA confirmed by NMR or other techniques should be stated, which is important for sensing properties. The particle morphology should be characterized by SEM or other methods, not only to check amorphous impurities but also because it can affect the sensing kinetics and processability.*

We appreciate the reviewer's thorough evaluation and valuable suggestions. In response, we revised our manuscript and Supplementary information as follows:

Changes made:

- We have revised Fig. 1b and Supplementary Figure 1, which includes a magnified view of the high-angle region to facilitate a more detailed inspection.

Fig. 1. (b) XRPD patterns of MOF-808-EDTA-Cu (blue) and MOF-808-EDTA-Cu (HCl) (red) with simulated XRPD pattern of MOF-808 (black)

Supplementary Figure 1. XRPD patterns of MOF-808 (black) and MOF-808-EDTA (red) with simulated XRPD pattern of MOF-808 (gray)

- We have revised the relevant sentences concerning the N₂ sorption isotherms in the manuscript. Additionally, we have revised Supplementary Figure 7 and included a discussion of the N₂ sorption isotherm of MOF-808, comparing it with previously reported data, in the revised Supplementary Information.

→ (Line 98) “Nitrogen sorption measurements of MOF-808-EDTA-Cu showed a decrease both in surface area and pore size of larger cavity (1118 m²/g and 0.81 nm) compared to that of the pristine MOF-808 (2065 m²/g and 1.28 nm), indicating that Cu chelated EDTA, the colorimetric center, exists in accessible internal pores of MOF-808 rather physical mixed (Fig. 1c and Supplementary Fig. 7).”

→ (Line 312) “The surface areas were calculated using BETSI, following the Rouquerol criteria 1-4.⁵⁸ Pore size distribution was calculated using Horvath-Kawazoe (HK) method.”

New reference :

58. Osterrieth, J. W. et al. How reproducible are surface areas calculated from the BET equation?. *Adv. Mater.* **34**, 2201502 (2022).

Supplementary Figure 7. N₂ adsorption isotherms of (a) MOF-808 and (b) MOF-808-EDTA-Cu. Red circles indicate the data points used for BET surface area calculations. The calculated BET areas for (c) MOF-808 and (d) MOF-808-EDTA-Cu, which were determined following the Rouquerol criteria 1-4 with BETSI program². Note that, the calculated BET area of MOF-808 is 2065 m²/g, which is consistent with previously reported values (1591~2424 m²/g)³⁻⁵. Pore size distribution graphs of (e) MOF-808 and (f) MOF-808-EDTA-Cu.

New references:

2. Osterrieth, J. W. et al. How reproducible are surface areas calculated from the BET equation?. *Adv. Mater.* **34**, 2201502 (2022).
3. Furukawa, H. et al. Water adsorption in porous metal–organic frameworks and related materials. *J. Am. Chem. Soc.* **136**, 4369-4381 (2014).
4. Peng, Y. et al. A versatile MOF-based trap for heavy metal ion capture and dispersion. *Nat. Commun.* **9**, 187-195 (2018).
5. Aunan, E. et al. Modulation of the Thermochemical Stability and Adsorptive Properties of MOF-808 by the Selection of Non-structural Ligands. *Chem. Mater.* **33**, 1471-1476 (2021).

- We have revised the NMR spectra in Supplementary Figure 2 as follows.

Supplementary Figure 2. ^1H NMR spectra of MOF-808 (black) and MOF-808-EDTA (red) in $\text{D}_2\text{SO}_4/\text{DMSO-d}_6$ solution. Based on the ^1H NMR spectrum of MOF-808-EDTA, the mole ratio of EDTA to 1,3,5-benzenetricarboxylic acid (BTC) is 0.89. Note that formic acid originates from MOF-808, which has the ideal structural formula $[\text{Zr}_6\text{O}_4(\text{OH})_4(\text{BTC})_2(\text{HCOO})_6]$.

- We have revised the manuscript and newly added Supplementary Figure 3 to help identify any amorphous impurities and to provide insights into how particle morphology may affect sensing kinetics and processability.

→ (Line 84) “The X-ray powder diffraction (XRPD) patterns and SEM image of MOF-808-EDTA-Cu confirmed that the framework structure and morphology was maintained even after post-synthetic treatments (Fig. 1b, Supplementary Fig. 1 and 3).”

Supplementary Figure 3. SEM images of MOF-808, MOF-808-EDTA, and MOF-808-EDTA-Cu.

- Lines 85 and 145: Here the amount of Cu^{2+} ions in MOF-808-EDTA-Cu is stated, which is important for sensing properties. However, the information on how this value is calculated based on ICP-AES measurement is missing. How is the coordination per EDTA calculated from the Cu weight fraction obtained by ICP-AES? Is the possibility of Cu^{2+} binding at different sites such as defects or SBUs considered?

Thank you for the valuable suggestions. We have included the ICP-AES results and a detailed description of the calculation method in Supplementary Tables 1 and 2. Additionally, we conducted new experiments to investigate the possibility of Cu^{2+} binding at different sites, such as defects or SBUs, within the MOF-808 structure. ICP-AES analysis of MOF-808-Cu, which shows that a negligible amount of Cu^{2+} is incorporated into the MOF-808 structure,

indicates that the majority of Cu^{2+} present in MOF-808-EDTA-Cu is due to EDTA chelation. The results of these experiments have been incorporated into the revised manuscript and supplementary information as detailed below:

Changes made:

- We have revised the sentences in revised manuscript.
 - (Line 86) “Inductively coupled plasma-atomic emission spectrometry (ICP-AES) ~ incorporated into MOF-808-EDTA-Cu (Supplementary Table 1 and 2).”
 - (Line 147) “Consequently, the EDTA-decorated MOF-808 ~ as confirmed by ICP-AES analysis (Supplementary Table 1).”
- We have added ICP-AES results as newly added Supplementary Tables 1 and 2 in revised supplementary information.

Supplementary Table 1. ICP-AES analysis results for MOF-808-Cu, MOF-808-EDTA-Cu, and MOF-808-EDTA-Cu-regen.

	Zr		Cu		Zr : Cu (molar ratio)
	mg/L	mmol/L	mg/L	mmol/L	
MOF-808-EDTA-Cu	78.457	0.860	13.193	0.208	1.000 : 0.242
MOF-808-EDTA-Cu-Regen (1 st)	17.093	0.187	2.265	0.036	1.000 : 0.193
MOF-808-EDTA-Cu-Regen (2 nd)	21.414	0.235	2.185	0.034	1.000 : 0.145
MOF-808-Cu ^a	84.745	0.929	0.079	0.001	1.000 : 0.001

^a MOF-808-Cu was prepared by immersing MOF-808 (not MOF-808-EDTA) in 100 mM Cu^{2+} solution for 24 h. ICP-AES analysis revealed a nearly negligible amount of Cu^{2+} in MOF-808-Cu, supporting the conclusion that the majority of Cu^{2+} present in MOF-808-EDTA-Cu is due to chelation by EDTA.

Supplementary Table 2. Calculation of the percentage of Cu^{2+} chelated by EDTA in MOF-808-EDTA-Cu.

	EDTA : BTC ^{a,b}	BTC ^a : Zr ^c	Zr : Cu ^d	EDTA : Cu
MOF-808-EDTA-Cu	0.89 : 1	1 : 3	3 : 0.72	0.89 : 0.72 (82%)

^a Abbreviation for 1,3,5-benzenetricarboxylic acid

^b The ratio of EDTA to BTC confirmed by NMR results (Supplementary Fig. 2)

^c The ratio of BTC to Zr in MOF-808 $[\text{Zr}_6\text{O}_4(\text{OH})_4(\text{BTC})_2(\text{HCOO})_6]$

^d The ratio of Zr to Cu determined by ICP-AES results (Supplementary Table 1)

4. Line 98: There is no information on how the surface areas are calculated. Are they calculated with the BET theory with robust criteria (ref. <http://doi.org/10.1002/adma.202201502>)?

We appreciate the reviewer’s valuable comment regarding the calculation of surface areas. In response to the reviewer's comment, we re-evaluated our surface area calculations to

enhance the reliability of our findings. We re-measured the N₂ adsorption isotherms and recalculated the surface areas using the BETSI software, adhering to the Rouquerol criteria 1–4 as recommended in the referenced article. As a result, the reported surface areas and pore size distributions for MOF-808-EDTA-Cu have been adjusted from the initial values (1128 m²/g and 0.79 nm) to revised values (1118 m²/g and 0.81 nm). Similarly, for MOF-808, the figures were updated from (2108 m²/g and 1.30 nm) to (2065 m²/g and 1.28 nm). Accordingly, we have updated the manuscript and supplementary information to reflect these changes.

Changes made:

- We have revised Fig. 1c and sentence in revised manuscript.

Fig. 1. (c) N₂ sorption isotherms of MOF-808 (black) and MOF-808-EDTA-Cu (blue) measured at 77 K.

→ (Line 98) “Nitrogen sorption measurements of MOF-808-EDTA-Cu showed a decrease both in surface area and pore size of larger cavity (1118 m²/g and 0.81 nm) compared to that of the pristine MOF-808 (2065 m²/g and 1.28 nm), indicating that Cu chelated EDTA, the colorimetric center, exists in accessible internal pores of MOF-808 rather physical mixed (Fig. 1c and Supplementary Fig. 7).”

→ (Line 310) “The surface areas were calculated using BETSI, following the Rouquerol criteria 1-4.⁵⁸ Pore size distribution was calculated using Horvath-Kawazoe (HK) method.”

New reference:

58. Osterrieth, J. W. et al. How reproducible are surface areas calculated from the BET equation?. *Adv. Mater.* **34**, 2201502 (2022).

- We have revised Supplementary Figure 7 to clearly illustrate the surface area calculations in the revised Supplementary Information.

Supplementary Figure 7. N₂ adsorption isotherms of (a) MOF-808 and (b) MOF-808-EDTA-Cu. Red squares indicate the data points used for BET surface area calculations. The calculated BET areas for (c) MOF-808 and (d) MOF-808-EDTA-Cu, which were determined following the Rouquerol criteria 1-4 with BETSI program². Note that the calculated BET area of MOF-808 is 2065 m²/g, which is consistent with previously reported values (1591~2424 m²/g)³⁻⁵. Pore size distribution graphs of (e) MOF-808 and (f) MOF-808-EDTA-Cu.

New references:

2. Osterrieth, J. W. et al. How reproducible are surface areas calculated from the BET equation?. *Adv. Mater.* **34**, 2201502 (2022).
 3. Furukawa, H. et al. Water adsorption in porous metal–organic frameworks and related materials. *J. Am. Chem. Soc.* **136**, 4369-4381 (2014).
 4. Peng, Y. et al. A versatile MOF-based trap for heavy metal ion capture and dispersion. *Nat. Commun.* **9**, 187-195 (2018).
 5. Aunan, E. et al. Modulation of the Thermochemical Stability and Adsorptive Properties of MOF-808 by the Selection of Non-structural Ligands. *Chem. Mater.* **33**, 1471-1476 (2021).
5. *Line 104, panel h: The "NH+" and "C-N" signals could be more accurately labeled as "R3NH+" and "R3N-Cu2+".*

We appreciate the reviewer’s insightful suggestion regarding the labeling of the signals. Following the reviewer’s suggestion, we revised the expression of manuscript and the label of Fig. 1h as below.

Changes made:

• We have revised the sentence and fig. 1h in revised manuscript.

→ (Line 167) “XPS spectra of the ~ N 1s of the protonated amine (R_3NH^+) and O 1s of the carboxylic acid in EDTA ~.”

Fig. 1 (h) N 1s XPS spectra of MOF-808 EDTA-Cu (top) and MOF-808 EDTA-Cu (HCl) (bottom).

6. *Line 131: How stable is the color change after switching the environment to the air?*

Thank you for the reviewer’s comments. Following the reviewer’s suggestions, we conducted additional experiments with extended exposure times. MOF-808-EDTA-Cu maintained its original cyan color even after 24 hours of exposure to potentially interfering air gases, confirming its stability and selectivity for acid gases only. We have updated the manuscript and supplementary information accordingly.

Changes made:

• We have revised the sentence in revised manuscript and Supplementary Figure 10 in revised supplementary information.

→ (Line 131) “Importantly, MOF-808-EDTA-Cu demonstrated a selective colorimetric response towards acid gas, maintaining its original cyan color unchanged after 24 hours of exposure to potentially interfering air gases such as N_2 , O_2 , and CO_2 , as well as variations in humidity and temperature within a substantial range (Supplementary Fig. 10).”

Supplementary Figure 10. Photographs of MOF-808-EDTA-Cu and exposed to N_2 , O_2 , CO_2 , 60 °C, 75% relative humidity air, 20 °C, 10% relative humidity air, 20 °C, 85% relative humidity air, 10 °C, 75% relative humidity air for 24 hours.

7. *Line 135: Here is the importance of the porous MOF structure demonstrated by comparison with pure $Cu(edta)(H_2O)$ solids. However, if only the porosity is important, it is possible the MOF-808 support can be replaced by other cheaper porous materials. For example, cobalt(II)*

chloride impregnated in silica gel or paper is known as a colorimetric sensor of water vapor. Does MOF-808 have advantages over these traditional porous materials?

We appreciate the reviewer's insightful comment and agree that other porous materials could potentially replace the MOF-808 support. Our research represents the first approach to specifically identify acid vapor using a single-domain sensor. To elucidate a decoding mechanism, a model material with precisely defined structures and specific active sites is required, criteria that MOF-808 fulfills as a practical model. Additionally, unlike traditional porous materials such as silica gel, MOF-808 offers enhanced density of active sites and superior porosity, key attributes for boosting the sensitivity and performance of our sensor. We strongly agree with the reviewer that exploring cheaper alternatives is essential, especially for the commercialization of acid decoding sensor. Thus, exploring cheaper alternatives will be an upcoming practical topic for further study, and we hope that the reviewer will agree with this conclusion.

8. *Line 143: Here the efficiency of regenerating the sensor material is discussed. Does this value depend on the amount of water used? Can this value be improved by using a buffer, base, or NH₃ vapor for effectively removing HCl?*

We appreciate the reviewer's insightful comments. Basic conditions can potentially improve HCl removal and sensor regeneration. To enhance regeneration efficiency, increasing the water volume from 10 mL to 780 mL for regenerating 10 mg of MOF-808-EDTA-Cu-HCl raised the pH of the solution from 2.5 to 4.3. However, this also diluted the copper concentration significantly, resulting in lower recovery efficiency (Reviewer only Table 2). When testing other HCl removal methods with basic chemicals (pH 8 buffer solution, triethylamine, and NH₃ vapor), only NH₃ vapor showed a notably better regeneration efficiency compared to water (Reviewer only Table 2). However, exposure to NH₃ vapor led to the formation of NH₄Cl, which could clog the sensor's pores and reduce repeatability (Reviewer only Figure 4). Given these considerations, we concluded that water is the simplest and most practical choice for sensor regeneration.

Reviewer only Table 2. ICP-AES analysis results for regenerated MOF-808-EDTA-Cu-HCl using water, pH 8 buffer, triethylamine, and NH₃ vapor.

	Zr		Cu		Zr : Cu (molar ratio)
	mg/L	mmol/L	mg/L	mmol/L	
MOF-808-EDTA-Cu	78.457	0.860	13.193	0.208	1.000 : 0.242
Water (10 mL)	17.093	0.187	2.265	0.036	1.000 : 0.193
Water (750 mL)	43.517	0.447	2.530	0.040	1.000 : 0.083
pH 8 buffer	17.222	0.189	1.248	0.019	1.000 : 0.101
Triethylamine	25.320	0.278	3.535	0.056	1.000 : 0.201
NH ₃ vapor	13.054	0.143	2.070	0.033	1.000 : 0.231

Reviewer only Figure 4. (a) Photographs of MOF-808-EDTA-Cu-HCl-Regen using deionized water, triethylamine, pH 8 buffer and NH_3 vapor. (b) XRPD patterns of MOF-808-EDTA-Cu-HCl-Regen using deionized water, triethylamine, pH 8 buffer and NH_3 vapor. Asterisk mark matches with NH_4Cl .

9. *Line 145: Here it is stated this sensor material shows reversible color changes. However, the experiment is done only for three cycles, and the UV-vis spectra show some irreversible changes. It would be better to state this partial reversibility quantitatively.*

We appreciate the reviewer's suggestion. During the three cycles of alternating exposure to HCl and water, the color of MOF-808-EDTA-Cu exhibited reversible cyan-yellow color variations observable to the naked eye, although it did show slight fading with each cycle. This observation is supported by ICP-AES results, which confirm that approximately 80% of Cu^{2+} ions were re-chelated by EDTA, as already mentioned in the original manuscript. In response to the reviewer's valuable feedback, we have revised the relevant sentences to better reflect this partial reversibility:

Changes made:

- We have revised the manuscript as follows:

- (Line 63) "Interestingly, the Cu-EDTA colorimetric center could be regenerated up to three times via a simple immersion in water."

→ "Interestingly, approximately 80% of the Cu-EDTA colorimetric centers could be regenerated through simple immersion in water, allowing the sensor to be reused up to three times."

- (Line 149) "Furthermore, during three cycles of alternating exposure to HCl and water, MOF-808-EDTA-Cu continued to exhibit reversible cyan-yellow color variations, highlighting the reusability of the sensor (Fig. 1e)."

→ "Furthermore, during three cycles of alternating exposure to HCl and water, the color of MOF-808-EDTA-Cu faded slightly with each cycle but still clearly exhibited reversible cyan-yellow color variations observable to the naked eye, highlighting the reusability of the sensor (Fig. 1e, Supplementary Figure 12 and Supplementary Table 1)."

10. *Line 176: Are the changes after exposure to HF, HBr, or HI gas also (partially) reversible?*

We appreciate the reviewer's comment regarding the reversibility of changes after exposure to another acids. Upon immersion in water, MOF-808-EDTA-Cu-HF, MOF-808-EDTA-Cu-HBr, and MOF-808-EDTA-Cu-HI were recovered as cyan powders through filtration, demonstrating partial reversibility (see Reviewer-only Figure 5). Note that, the XRPD patterns of HF exposed and after immersion in water samples shows higher backgrounds and noises than other acid exposed and regenerated samples, implying partial decomposition of the MOF-808 framework structure by HF, which is reasonable for Zr-MOFs.

Reviewer only Figure 5. (a) Photographs of MOF-808-EDTA-Cu-HF-Regen, MOF-808-EDTA-Cu-HBr-Regen and MOF-808-EDTA-Cu-HI-Regen using deionized water. (b) XRPD patterns of MOF-808-EDTA-Cu-HI-Regen (light orange), MOF-808-EDTA-Cu-HI (orange), MOF-808-EDTA-Cu-HBr-Regen (light purple), MOF-808-EDTA-Cu-HBr (purple), MOF-808-EDTA-Cu-HF-Regen (light green) and MOF-808-EDTA-Cu-HF (green) with simulated XRPD patterns of MOF-808 (black).

11. Line 183: Here the formation of "I₂(aq)" is discussed, although there is no aqueous medium in this experiment. Is this a mistake of I₂ adsorbed in the material? How is the product identified as I₂? Does the color fade over time by the evaporation of I₂?

We apologize for the confusion caused by the reference to "I₂(aq)" in our manuscript. The appropriate term should be I₃⁻, which forms through the reaction of I₂ with I⁻ in the presence of Cu²⁺ ions. The relevant redox reactions are as follows:

In our experiments involving the MOF-808-EDTA-Cu system and HI exposure, the formation of white CuI and brown I₃⁻ was confirmed through PXRD and UV-vis-NIR spectra analysis, respectively (Supplementary Figure 18). Although our experiments occur in a humid atmosphere where MOF-808 is known to absorb atmospheric moisture (Furukawa, H. *et al. J. Am. Chem. Soc.* **136**, 4369-4381 (2014)), describing the medium as aqueous would be inaccurate. Accordingly, the term "I₂(aq)" has been corrected to "I₃⁻" to prevent any misunderstanding.

Additionally, Reviewer only Figure 6 demonstrates that the initial brown color of MOF-808-EDTA-Cu(HI) slowly faded to light brown over 24 hours when exposed to air, implying decomposition of I₃⁻.

Changes made:

- We have revised the manuscript as follows:

→ (Line 186) "Furthermore, exposure to HI resulted in the formation of white CuI(s) and brown I₃⁻, as confirmed by PXRD and UV-vis-NIR spectra, suggesting that the brown color of the HI-exposed MOF-808-EDTA-Cu originated from I₃⁻ rather than CuI(s) (Supplementary Fig. 18)."

Supplementary Figure 18. (a) XRPD patterns of MOF-808-EDTA-Cu (HI) (orange) with simulated XRPD patterns of CuI (gray) and MOF-808 (black), (b) Diffuse reflectance UV-vis-NIR spectra of MOF-808-EDTA-Cu (HI) (orange). (c) Diffuse reflectance UV-vis-NIR spectra of MOF-808-EDTA-Cu (HI) in the 200-600 nm wavelength range. (d) Reported UV-vis spectrum of triiodide in ED-600/iodoethane/I₂ electrolyte showing absorption peaks of I₃⁻ at 298 and 367 nm. Reprinted with permission from ref 7. Copyright 2011 The Royal Society of Chemistry.

New reference :

7. Apostolopoulou, A., Margalias, A. & Stathatos, E. Functional quasi-solid-state electrolytes for dye sensitized solar cells prepared by amine alkylation reactions. *RSC Adv.* **5**, 58307-58315 (2015).

Reviewer only Figure 6. Photographs of MOF-808-EDTA-Cu and MOF-808-Cu-HI in air over time (0 min, 30 min, 1 h, 2 h, 4 h, and 24 h).

12. Line 189: Is there any information about the acidochromism mechanism of MOF-808-EDTA-Fe? For example, do the UV-vis absorption peaks match the LMCT transitions of [FeX₄]-?

We thank the reviewer's insightful question. The UV-vis-NIR spectra of MOF-808-EDTA-Fe after exposure to HCl and HBr were consistent with those of FeCl₄⁻ and FeBr₄⁻, respectively⁵³, implying the formation of FeCl₄⁻ and FeBr₄⁻ within the MOF sensor. Thus, we revised manuscript and supplementary figure 19 as follows.

Changes made:

- We have added the sentence in revised manuscript.

→ (Line 190) "To further explore our methodology, we prepared Fe³⁺-chelated MOF-808-EDTA (referred to as MOF-808-EDTA-Fe) as a colorimetric sensor, which exhibited a distinct color change from ivory to yellow and orange upon exposure to HCl and HBr, respectively, confirming the formation of FeCl₄⁻ and FeBr₄⁻ within the sensor after acid exposure⁵³ (Fig. 2b and Supplementary Fig. 19)."

New reference:

53. Döbbelin, M. et al. Synthesis of paramagnetic polymers using ionic liquid chemistry. *Polymer Chemistry*, **2**, 1275-1278 (2011).

- We have revised Supplementary Figure 19.

New Reference :

8. Döbbelin, M. et al. Synthesis of paramagnetic polymers using ionic liquid chemistry. *Polym. Chem.* **2**, 1275-1278 (2011).

13. Line 197: Are the cobalt ions in this material confirmed to be Co^{2+} but not Co^{3+} ?

Thank you for the reviewer's comments regarding the confirmation of cobalt ions. Cobalt(II) nitrate was used in the preparation of MOF-808-EDTA-Cu/Co. The UV-vis-NIR spectra for both MOF-808-EDTA-Co and MOF-808-EDTA-Cu/Co exhibit absorption peaks around 494 nm, consistent with the characteristic Co(II) EDTA peak at 490 nm and distinct from the Co(III) EDTA peak observed at 535 nm (Reviewer only Figure 7). These findings confirm the cobalt ions are in the Co^{2+} oxidation state.

Reviewer only Figure 7. (a) Diffuse reflectance UV-vis-NIR spectra of MOF-808-EDTA-Cu (black), MOF-808-EDTA-Co (red), and MOF-808-EDTA-Cu/Co (blue) (b) Diffuse reflectance UV-vis-NIR spectra of MOF-808-EDTA-Cu (black), MOF-808-EDTA-Co (red), and MOF-808-EDTA-Cu/Co (blue) in the 400-600 nm wavelength range. (c) Reported UV-vis spectra of Co(II) EDTA (dotted line) and Co(III) EDTA (solid line) in the 400-600 nm wavelength range. Reprinted with permission from Reviewer only ref 1. Copyright 2015 IJASBT, Permits unrestricted use under the CC-BY-NC 4.0 License.

Reviewer only reference 1: Paraneiswaran, A., Shukla, S. K., Sathyaseelan, V. S. & Rao, T. S. A spectrophotometric method for the determination Co-EDTA complexes. *Int. J. Appl. Sci. Biotechnol*, **3**, 584-587 (2015).

14. Line 201: Is there any information about the chromism mechanism of MOF-808-EDTA-Cu/Co by nitric acid and TFA? Was the used nitric acid pure and free from NO_2 ?

We thank the reviewer for the insightful question. As shown in the FT-IR spectra (Reviewer-only Figure 8), exposure of MOF-808-EDTA-Cu/Co to TFA and nitric acid led to protonation of the carboxylate groups in Cu/Co-EDTA, indicated by a decrease in the peak intensity at ca. 1563 cm^{-1} , corresponding to the $\nu_{\text{as,COO}^-}$ of EDTA, and the appearance of a new peak at ca. 1713 cm^{-1} , attributed to the $\nu_{\text{C=O}}$ of carboxylic acids.

Additionally, new peaks were observed at 1651 cm^{-1} in the TFA-exposed sample and $1296, 1120, 793 \text{ cm}^{-1}$ in the nitric acid-exposed sample, confirming the presence of deprotonated

trifluoroacetate (CF_3COO^-) and nitrate (NO_3^-), respectively, both of which possible to coordinate with the released metal ions. Photographs and UV-vis-NIR spectra further support that the coordination environment of Cu^{2+} and Co^{2+} changes upon acid exposure (Reviewer-only Figure 9). Although the exact structure of the resultant metal complexes cannot be determined with the current data, the IR and UV spectra suggest an acid-triggered color change mechanism.

Regarding the reviewer's concern about the nitric acid used in our experiments, we would like to clarify that nitric acid containing NO_2 appears as a light brown liquid, typically seen in older samples. However, we used freshly opened nitric acid, which is colorless and transparent, ensuring that the acid was pure and free from NO_2 , as shown in Reviewer-only Figure10.

Reviewer only Figure 8. (a) FT-IR spectra of MOF-808-EDTA-Cu/Co (black) and MOF-808-EDTA-Cu/Co-HNO₃ (blue). (b) FT-IR spectra of MOF-808-EDTA-Cu/Co (black) and MOF-808-EDTA-Cu/Co-TFA (red).

Reviewer only Figure 9. (a) Photographs of MOF-808-EDTA-Cu/Co-TFA, MOF-808-EDTA-Cu/Co-HNO₃ and MOF-808-EDTA-Cu/Co (b) Diffuse reflectance UV-vis-NIR spectra of MOF-808-EDTA-Cu/Co-TFA (pink), MOF-808-EDTA-Cu/Co-HNO₃ (purple), and MOF-808-EDTA-Cu/Co (black)

Reviewer only Figure 10. Photograph of Nitric acid used for nitric acid-exposed experiments.

15. Line 209: Is the fabricated sensor device reusable as shown in the case of the powder form? Is the color retained after exposure to air?

We thank the reviewer for bringing this issue to our attention. Like its powder form, the sensor device is reusable and can be regenerated by washing with water. However, we observe slight fading with each cycle, as detailed in Supplementary Figure 22. Furthermore, the MOF-808-EDTA-Cu sensor device consistently retained its original cyan color after exposure to 85% RH humid air for 24 hours, confirming its selective colorimetric response to acid gases only. These observations have been incorporated into the revised manuscript to further elucidate the device's capabilities and stability under varying environmental conditions.

Changes made:

- We have newly added the sentence about sensor device reusability and newly added Supplementary Figure 22.

→ (Line 218) “When exposed to HCl vapor evaporating from a concentrated HCl solution (approximately 15,500 ppm)⁵⁶⁻⁵⁷, the MOF-808-EDTA-Cu portable sensor underwent a distinct color

shift from cyan to yellow. This cyan-yellow color variation persisted over three cycles of alternating exposure to HCl and water (Supplementary Fig. 22). The distinct cyan-to-yellow color change was detectable by the camera sensor and translated into RGB channel values (Fig. 3b), allowing the quantification of the color changes and 24-hour real-time monitoring.”

→ (Line 390) “To regenerate the MOF-808-EDTA-Metal portable sensor, it was completely submerged in a glass dish containing 10 mL of water, removed, and allowed to vacuum dry.”

Supplementary Figure 22. Photographs of MOF-808-EDTA-Cu portable sensor during a series of hydrochloric acid vapor exposure and regeneration cycles.

• To emphasize the sensor's air stability, we extended the air exposure time in Fig. 3d from 5 minutes to 24 hours in revised manuscript.

Fig. 3 (d) Time-dependent RGB curves of MOF-808-EDTA-Cu portable sensor exposed to 85% RH water vapor for 24 hours. Inset: Photographs of MOF-808-EDTA-Cu portable sensor under 85% RH water vapor exposure.

16. *Line 280: There is no information on the used materials and chemicals such as purity and supplier, which are important for replication of this work.*

We appreciate the reviewer’s comment about the information on materials and chemicals. In response to the comment, we have added information on the used materials and chemicals in the method section.

Changes made:

• We have added the sentences as follows;

(Line 293) “1,3,5-Benzentricarboxylic acid (95%), sulfuric acid-d₂ (96-98%), hydrofluoric acid (48%), copper(II) nitrate trihydrate (99-104%), iron(III) nitrate nonahydrate (98%), trifluoroacetic acid (99%), poly(vinylidene fluoride) (average MW : ~ 534,000), copper(II) acetate monohydrate (98%), sodium chloride (99%), potassium chloride (99%), copper(I) iodide (98%) were purchased from Sigma-Aldrich. Zirconium dichloride oxide octahydrate (98%), formic acid (99%), cobalt(II) nitrate hexahydrate (98-102%), hydrobromic acid (48%), hydroiodic acid (55-58%) were purchased from Thermo Fisher

Scientific. Dimethyl sulfoxide-d₆ (99.9%) was purchased from Cambridge Isotope Laboratories. Nitric acid (60%), N,N-dimethylformamide (99.5%), acetone (99.5%), ethylenediaminetetraacetic acid disodium salt (99%), hydrochloric acid (35-37%) were purchased from DAEJUNG chemicals. Copper(II) fluoride (98%) was purchased from Tokyo Chemical Industry.”

17. *Line 285: The procedures for digesting solid samples for solution NMR spectroscopy are missing.*

We apologize for the omission regarding the procedures for NMR sampling. In response to the comment, we have added the procedures for digesting solid samples for solution NMR spectroscopy as follows:

Changes made:

- We have added the NMR sample preparation procedures in revised manuscript as follows;
→ (Line 307) “For NMR sample preparation, 0.005 g of samples were digested with 20 μ L of D₂SO₄, followed by dissolution in 600 μ L of DMSO-d₆.”

18. *Line 307 and others: The ¹H NMR chemical shifts are reported as values in "DMSO-d6", which is likely to be a mistake of D2SO4/DMSO-d6 with a certain ratio.*

We apologize for our carelessness. We have revised from “DMSO-d₆” to “D₂SO₄/DMSO-d₆” at line 331, 337 and caption of Supplementary Figure 2.

Changes made:

- We have revised the sentence as follows;
→ (Line 331) “The MOF-808 was ~, ¹H-NMR (D₂SO₄/DMSO-d₆): δ 8.56 (s, 3H) ~.”
→ (Line 337) “¹H-NMR (D₂SO₄/DMSO-d₆): δ 8.56 (s, 3H), δ 4.10 (s, 8H), δ 3.60 (s, 4H), ~.”
→ (Caption of Supplementary Figure 2) “¹H NMR spectra of MOF-808 (black) and MOF-808-EDTA (red) in D₂SO₄/DMSO-d₆ solution. Based on the ¹H NMR spectrum of MOF-808-EDTA, the mole ratio of EDTA to 1,3,5-benzenetricarboxylic acid (BTC) is 0.89. Note that formic acid originates from MOF-808, which has the ideal structural formula [Zr₆O₄(OH)₄(BTC)₂(HCOO)₆].”

19. *Line 328: The procedures for acid vapor detection are described only for HCl, and those for other acids are missing.*

Thank you for pointing out the missing procedures for acid vapor detection. Following the comment, we have added the procedures for acid vapor detection for other acids.

Changes made:

- We have revised “Acid Vapor Detection and Regeneration of MOF-808-EDTA-metal” section in Methods in revised manuscript.

→ (Line 353) “3 cm glass dish containing 0.010 g of MOF-808-EDTA-Metal was placed inside a 5 cm glass dish with 2 mL of concentrated HCl solution, ensuring no direct contact between the MOF-808-EDTA-metal and the acid solution. The 5 cm dish was covered with a 7 cm glass dish to detect vaporized acid. For HF, HBr, HI, HNO₃, and CF₃COOH vapor detection experiment, 2 mL of respective concentrated acid solutions were used instead of concentrated HCl solution. Notably, HF solution was handled using polystyrene petri dish to avoid contact with glass.

For different concentration of HCl vapor detection, varying concentrations of HCl vapor were prepared using HCl solutions with different wt% based on a previously reported method⁵⁶⁻⁵⁷; 15460 ppm, 740 ppm, 590 ppm, 300 ppm and 120 ppm HCl vapor were prepared using 37.1 wt%, 24.7 wt%, 22.0 wt% 21.6 wt% and 18.5 wt% HCl solutions, respectively.”

20. SI Line 33: There is no information on how the pore size distributions are calculated.

Thank you for highlighting the lack of information on calculations for pore size distributions. Following the comment, we have added the information on how the pore size distributions were calculated in the caption of Supplementary Figure 6.

Changes made:

- We have newly added the sentence in revised manuscript.

→ (Line 311) “Pore size distribution was calculated using Horvath-Kawazoe (HK) method.”

21. SI Line 74 panel b: The PXRD pattern after HF exposure shows a higher background and noise. Does this indicate the decomposition of the MOF-808 framework by HF, which is reasonable for Zr-MOFs?

We thank the reviewer for bringing this issue to our attention. The PXRD pattern after 30 minutes of HF exposure shows a higher background and increased noise, suggesting partial decomposition of the MOF-808 framework. This observation is further supported by additional experiments, where prolonged HF exposure (4 hours, 8 times longer) resulted in the complete decomposition of MOF-808, as shown in Reviewer Figure 11. This aligns with the known behavior of Zr-MOFs, which typically degrade upon HF exposure. To address this, we have included a discussion on the decomposition of MOF-808 in the revised Supplementary Information (SI Line 74, panel b) for clarity.

Reviewer Only Figure 11. XRPD patterns of MOF-808-EDTA-Cu (blue), MOF-808-EDTA-Cu (HF, 30min) (green), and MOF-808-EDTA-Cu (HF, 4 h) (blue green) with simulated XRPD pattern of MOF-808 (gray).

Changes made:

- We have revised the caption in Supplementary Figure 16 in revised SI.

→ (Caption of Supplementary Figure 16) “(a) Diffuse reflectance UV-vis-NIR spectra of MOF-808-EDTA-Cu (HF) (dark green), $\text{CuF}_2 \cdot 2\text{H}_2\text{O}$ (black), and MOF-808-EDTA-Cu (blue). (b) XRPD patterns of MOF-808-EDTA-Cu (HF) (dark green) with simulated XRPD pattern of MOF-808 (black). Note that, high background intensity in XRPD pattern of MOF-808-EDTA-Cu (HF) implies that partial decomposition of MOF-808 structure.”

- 22.** *General: It is recommended to attach adsorption data in the adsorption information file (AIF) format as supplementary materials or upload them to a data repository for accurate inspection and usage by readers.*

We appreciate the reviewer's suggestion. In accordance with the reviewer's recommendation, we have included the AIF files for the adsorption data of MOF-808 and MOF-808-EDTA-Cu presented in our paper in the supplementary data 1 and 2, respectively.

Responses to the Comments of Reviewer #1

(Reviewer's Comments) *In the revised version, most issues have been addressed. However, further revisions are highly recommended to enhance the manuscript before it is published in Nature Communications. The details are as follows.*

(Author's Response) We appreciate the reviewer's constructive review and feedback to improve the paper. We have addressed in detail each of the points raised by the reviewer in the attached responses. We hope the reviewer finds these satisfactory and thank the reviewer once more for the time and effort invested in the review process. Please find below our point-by-point response to the reviewer's comments.

1. *To emphasize the significance of the designed sensor utilizing MOF-808-EDTA-Cu, it is crucial to compare it with previously reported acid vapor-sensing materials. Please include these comparisons. Although the authors added four new references in revised manuscript, the significance of the designed sensor utilizing MOF-808-EDTA-Cu is still not emphasized. It is strongly recommended to include a comparison table to better highlight the sensing performance of the designed sensor.*

Thank you for the reviewer's valuable suggestions to include a comparison table, which will emphasize the performance of our sensor to the readers. In accordance with the reviewer's recommendations, we have included the comparison table as Supplementary Table 1 in the revised manuscript.

Changes made:

- We have revised sentences in the revised manuscript and have newly added supplementary table 1 in revised supplementary information.

→ (Line 37) "Recently, several colorimetric sensors for the detection of acid vapors have been reported utilizing organic dyes¹⁸⁻²¹, polymers²²⁻²³, molecular metal complexes²⁴⁻²⁶, covalent organic frameworks (COFs)²⁷⁻²⁹, and metal-organic frameworks (MOFs)^{8,30}."

→ (Line 39) "However, most studies have focused on detecting single HCl vapors by relying on a protonation mechanism, which cannot differentiate between various acid vapors (Supplementary Table 1)."

New references:

21. Li, K. et al. Solvatochromism, acidochromism and aggregation-induced emission of propeller-shaper spiroborates. *Dalton Trans.* **47**, 15002-15008 (2018).

21. Subodh, Prakash, K. & Masram, D. T. Chromogenic covalent organic polymer-based microspheres as solid-state gas sensor. *J. Mater. Chem. C.* **8**, 9201-9204 (2020).

22. Subodh, Prakash, K. & Masram, D. T. A reversible chromogenic covalent organic polymer for gas sensing applications. *Dalton Trans.* **49**, 1007-1010 (2020)

Supplementary Table 1. Summary of representative acid vapor sensor studies.

Sensor Materials	Color Change Mechanism	Detected Acid Vapors	Decoding Ability	Ref.
MOF-808-EDTA-Metal	Proton-triggered metal-anion coordination	HCl, HF, HBr, HI, HNO ₃ , TFA	Yes	This work
Organic dyes				
1,3,3-Trimethylindolino-6'-nitrobenzopyrlospiran	Protonation	HCl	No	9, 10
8-methoxy-1',3',3',-trimethyl-6-nitrospiro[chromene-2,2'-indoline]	Protonation	HCl	No	11
Sborepy3	Protonation	HCl	No	12

Sborepy6				
Polymer				
TATF-COM	Protonation	HCl	No	13
CC-TATP-COP	Protonation	HCl	No	14
Molecular Metal Complexes				
[ZnHLCI ₂].THF	Phase change	HCl	No	15
[ZnHLCI ₂].DMF				
[Cd ₂ L ₂ Cl ₄].CH ₃ CN	Phase change	HCl	No	16
ZnLCl ₂				
CdHL ₂ Cl ₂	Protonation	HCl	No	17
Covalent Organic Frameworks (COFs)				
PhotoPAN	Protonation	HCl	No	18
Per-Py COF	Protonation	TFA	No	19
Per-N COF				
Py-TT COF				
CZ-DHZ-COF	Protonation	HCl	No	20
Metal Organic Frameworks (MOFs)				
Zn ₁₄ (TCPE) ₅ (NO ₂) ₄ (O) ₆	HCl adsorption,	HCl	No	21
Co ₁₄ (TCPE) ₅ (NO ₂) ₄ (O) ₆	dipole effect			
CSMCRI-6	Protonation	HCl	No	22
Colorimetric Sensor Arrays				
36 different responsive pigments (Methyl Red, Fluorescein, Reichart's Dye, Pb(OAc) ₂ , etc.)		HF, HCl, HNO ₃	Yes	23

New supplementary references:

- Nam, Y. S. et al. Photochromic spiropyran-embedded PDMS for highly sensitive and tunable optochemical gas sensing. *Chem. Commun.* **50**, 4251-4254 (2014).
- Guo, J., Wei, X., Fang, X., Shan, R. & Zhang, X. A rapid acid vapor detector based on spiropyran-polymer composite. *Sens. Actuators B Chem.* **347**, 130623 (2021).
- Genovese, M. E. et al. Light responsive silk nanofibers: an optochemical platform for environmental applications. *ACS Appl. Mater. Interfaces* **9**, 40707-40715 (2017).
- Li, K. et al. Solvatochromism, acidochromism and aggregation-induced emission of propeller-shaper spiroporates. *Dalton Trans.* **47**, 15002-15008 (2018).
- Subodh, Prakash, K. & Masram, D. T. Chromogenic covalent organic polymer-based microspheres as solid-state gas sensor. *J. Mater. Chem. C.* **8**, 9201-9204 (2020).
- Subodh, Prakash, K. & Masram, D. T. A reversible chromogenic covalent organic polymer for gas sensing applications. *Dalton Trans.* **49**, 1007-1010 (2020)
- Liang, Q.-F., Zheng, H.-W., Yang D.-D. & Zheng, X.-J. Zn(II) complexes based on a Schiff base: mechanochromism- and solvent molecule-dependent acidochromism. *Cryst. Growth Des.* **22**, 3924-3931 (2022).
- Yang, D.-D. et al., Multi-stimuli responsive behavior of two Schiff base complexes with high contrast multicolor switching and wearable applications for rapid detection of HCl and NH₃ vapor. *Dyes and Pigments* **212**, 111149 (2023).
- Liang, Q.-F., Zheng, H.-W., Yang, D.-D. & Zheng, X.-J. A triphenylamine derivative and its Cd(II) complex with high-contrast mechanochromic luminescence and vapochromism. *CrystEngComm* **24**, 543-551 (2022).
- Kundu, P. K., Olsen, G. L., Kiss, V. & Klajn, R. Nanoporous frameworks exhibiting multiple stimuli responsiveness. *Nat. Commun.* **5**, 3588 (2014)
- Ascherl, L. et al. Perylene-based covalent organic frameworks for acid vapor sensing. *J. Am. Chem. Soc.* **141**, 15693-15699 (2019).
- Gong, W. et al. Dual-function fluorescent hydrazone-linked covalent organic frameworks for acid vapor sensing and iron (iii) ion sensing. *J. Mater. Chem. C* **10**, 3553-3559 (2022).
- Zhu, Z. H., Ni, Z., Zou, H. H., Feng, G. & Tang, B. Z. Smart metal-organic frameworks with reversible luminescence/magnetic switch behavior for HCl vapor detection. *Adv. Func. Mater.* **31**, 2106925 (2021).
- Goswami, R., Das, S., Seal, N., Pathak, B. & Neogi, S. High-performance water harvester framework for triphasic and synchronous detection of assorted organotoxins with site-memory-reliant security encryption via pH-triggered fluoroswitching. *ACS Appl. Mater. Interfaces* **13**, 34012-34026.

(2021).

23. Feng, L. et al. Colorimetric sensor array for determination and identification of toxic industrial chemicals. *Anal. Chem.* **82**, 9433-9440 (2010).

2. In Fig. 1e, MOF-808-EDTA-Cu demonstrated reversibility over three cycles. Why specifically three cycles or more? The findings are intriguing, as they reveal a decrease in copper content during the reversibility process. It is recommended to include this evidence in the revised manuscript. It would provide valuable insight and offer a useful approach for exploring the reversibility of sensing materials.

We appreciate the reviewer's insightful suggestion to include supplementary data regarding copper content during the reversibility process, as it could provide valuable insight to readers. In response to the reviewer's feedback, we have incorporated the data into the revised manuscript, as demonstrated below.

Changes made:

• We have revised sentences in the revised manuscript and revised supplementary figure 13 and supplementary table 2 in revised supplementary information.

→ (Line 63) "Interestingly, approximately 80% of the Cu-EDTA colorimetric center could be regenerated through simple immersion in water, allowing the sensor to be reused up to three times while retaining half of its initial Cu content."

→ (Line 148) "Consequently, the EDTA-decorated MOF-808 demonstrated an ability to re-chelate approximately 80% of Cu^{2+} ions during the regeneration process, as confirmed by ICP-AES analysis (Supplementary Table 2). Furthermore, during of alternating exposure to HCl and water, the color of MOF-808-EDTA-Cu faded slightly with each cycle but still clearly exhibited reversible cyan-yellow color variations observable to the naked eye, highlighting the reusability of the sensor (Fig. 1e, Supplementary Fig. 13 and Supplementary Table 2)."

Supplementary Figure 13. (a) Photographs of MOF-808-EDTA-Cu during a series of hydrochloric acid vapor exposure and regeneration cycles. (b) XRPD patterns of MOF-808-EDTA-Cu at each stage during five cycles of alternating exposure to HCl vapor and water. (c) Diffuse reflectance UV-vis-NIR spectra of MOF-808-EDTA-Cu, 1st HCl exposed MOF-808-EDTA-Cu, 1st regenerated MOF-808-EDTA-Cu, 2nd HCl exposed MOF-808-EDTA-Cu, 4th regenerated MOF-808-EDTA-Cu, and 5th HCl exposed MOF-808-EDTA-Cu.

Supplementary Table 2. ICP AES analysis results for MOF-808-Cu, MOF-808-EDTA-Cu, and MOF-808-EDTA-Cu-regen.

Zr	Cu	Zr : Cu
----	----	---------

	mg/L	mmol/L	mg/L	mmol/L	(molar ratio)
MOF-808-EDTA-Cu	78.457	0.860	13.193	0.208	1.000 : 0.242
MOF-808-EDTA-Cu-Regen (1 st)	17.093	0.187	2.265	0.036	1.000 : 0.193
MOF-808-EDTA-Cu-Regen (2 nd)	21.414	0.235	2.185	0.034	1.000 : 0.145
MOF-808-EDTA-Cu-Regen (3 rd)	21.163	0.232	1.873	0.029	1.000 : 0.125
MOF-808-EDTA-Cu-Regen (4 th)	13.761	0.151	0.650	0.010	1.000 : 0.066
MOF-808-Cu ^a	84.745	0.929	0.079	0.001	1.000 : 0.001

^a MOF-808 (without EDTA) was immersed in 10 mL of 0.1 M aqueous Cu²⁺ solution for 24 hours. ICP-AES analysis showed a nearly negligible amount of Cu²⁺ in MOF-808-Cu, indicating that the majority of Cu²⁺ present in MOF-808-EDTA-Cu is due to the chelation effect of EDTA.

3. *It is not clear why PVDF was chosen as the polymer matrix and whether the mass of PVDF affects the sensing performance of the MOF/PVDF sensor. Please provide further explanations. Although the PVDF content was optimized, the specific optimal amount is not clearly indicated in the revised manuscript. Please include this information.*

Thank you for pointing out the need to clarify the optimal PVDF amounts in the revised manuscript. Following the reviewer's comment, we have included this information in the revised manuscript and supplementary information.

Changes made:

- We have revised sentence in the revised manuscript.

→ (Line 218) "The MOF sensor-based ink with optimized PVDF amounts of 20wt% was applied to various substrates, including foil, paper, fabric, and glass (Supplementary Figs. 20 and 21)."

→ (Line 254) "In experiments with hydrohalic acid vapors, including HF, HBr, and HI, the color change of the MOF-808-EDTA-Cu portable sensor with 20wt% PVDF was not complete until 25 min, however, changes detectable by the naked eye appeared within 10 min (Supplementary Fig. 23)."

→ (Line 375) "Then, 0.6 mL of a DMF solution containing 0.012 g of PVDF ($M_w \sim 534,000$), yielding a PVDF concentration of 20wt%, was added to the MOF suspension."

→ (Line 384) "The MOF-808-EDTA-Metal portable sensor was prepared by coating 200 μ L of MOF sensor-based ink containing 20wt% PVDF onto a 1.8 cm x 1.8 cm cover glass."

→ (Caption of Fig. 3) "(b) Time-dependent RGB curves of MOF-808-EDTA-Cu portable sensor with 20wt% PVDF ~. Inset: Photographs of MOF-808-EDTA-Cu portable sensor with 20wt% PVDF ~. (d) ~ MOF-808-EDTA-Cu portable sensor with 20wt% PVDF ~. Inset: Photographs of MOF-808-EDTA-Cu portable sensor with 20wt% PVDF ~. (e) ~ MOF-808-EDTA-Cu portable sensors with 20wt% PVDF ~. Inset: Photographs of the MOF-808-EDTA-Cu portable sensor with 20wt% PVDF ~."

→ (Caption of Supplementary Figure 22) "Photographs of MOF-808-EDTA-Cu ink containing 20wt% PVDF applied to various substrates"

→ (Caption of Supplementary Figure 23) "Photographs of portable acid vapor decoding sensors processed on cover glass with varying PVDF amounts and after exposure to 15,500 ppm HCl vapor. The sensor with 20wt% PVDF, tailored for optimal performance, demonstrated the best results, while a low PVDF mass ratio of 10wt% resulted in poor film formation, and a high PVDF mass ratio of 40wt% led to a diminished cyan-to-yellow transition within the same exposure timeframe."

→ (Caption of Supplementary Figure 25) "Time-dependent RGB curves of MOF-808-EDTA-Cu portable sensors with 20wt% PVDF exposed to (a) HF, (b) HBr, and (c) HI vapor. (Inset) Photographs of MOF-808-EDTA-Cu portable sensors with 20wt% PVDF under acid vapor exposures."

Responses to the Comments of Reviewer #2

(Reviewer's Comments) *I am satisfied with the authors' revisions and responses to the comments provided. I recommend accepting this manuscript for publication in Nature Communications in its current form.*

(Author's Response) We sincerely thank the reviewer for their constructive feedback and for recognizing our efforts in revising the manuscript. We are pleased that the revised manuscript meets the reviewer's expectations. We deeply appreciate the reviewer's recommendation for publication and their valuable insights throughout the review process.

Responses to the Comments of Reviewer #3

(Reviewer's Comments) *The authors revised the manuscript thoroughly, and they addressed most of the points I and the other reviewers raised. This work is recommended for publication, given the following points are addressed.*

(Author's Response) We appreciate the time and effort the reviewer dedicated to reviewing our work, and for the reviewer's valuable feedback, which has significantly enhanced the quality and clarity of our manuscript. We have carefully addressed the points the reviewer raised. We hope the revised version meets the reviewer's expectations. Please find our point-by-point response to the reviewer's comments, below.

1. *My previous question, "10. Are the changes after exposure to HF, HBr, or HI gas also (partially) reversible?" was answered properly, but the manuscript was not revised accordingly. Since it is an advantage of this material that it can reversibly detect these gases, I believe it is worth noting.*

We sincerely thank the reviewers for their valuable suggestions to include data demonstrating the reversibility of MOF-808-EDTA-Cu upon exposure to HF, HBr, and HI gases, which further emphasizes the importance of our material. Following the reviewer's recommendation, we have revised the manuscript and supplementary information as follow.

Changes made:

- We have newly added Supplementary Fig. 20 and sentence in revised manuscript.
- (Line 188) "In addition, immersion of the acid-exposed MOF-808-EDTA-Cu in water restored its cyan color, thereby enabling recovery as a cyan powder through filtration (Supplementary Fig. 20)."

Supplementary Figure 20. (a) Photographs of MOF-808-EDTA-Cu-HF-Regen, MOF-808-EDTA-Cu-HBr-Regen, and MOF-808-EDTA-Cu-HI-Regen. (b) XRPD patterns of MOF-808-EDTA-Cu-HI-Regen (light orange), MOF-808-EDTA-Cu-HI (orange), MOF-808-EDTA-Cu-HBr-Regen (light purple), MOF-808-EDTA-Cu-HBr (purple), MOF-808-EDTA-Cu-HF-Regen (light green) and MOF-808-EDTA-Cu-HF (green) with simulated XRPD patterns of MOF-808 (black).

2. In response to my question 20, the authors described that the pore size distribution had been calculated using the Horvath–Kawazoe (HK) method. However, this method is obsolete and specific to graphitic slit pores in microporous carbons, which do not resemble the pores of MOF-808. This choice of method can be the reason that the pore size distribution in Figure S7 does not match the crystal structure or that in reference S5. The pore calculation should be conducted instead with an NLDFT or QSDFT method, which is more universal, accurate, and used in reference S5. Since the isotherms are recorded with an Autosorb sorptometer, the software ASiQwin can be used for such analysis.

We sincerely appreciate the reviewer's valuable comment on the pore size distribution analysis. In accordance with the reviewers' suggestions, we have re-performed the pore size distribution analysis using both the HK and QSDFT methods, confirming that the QSDFT method provides a more accurate approximation of the real structure of MOF-808 compared to the HK method. To ensure a more precise and comprehensive analysis, we have modified Supplementary Figure 8 to reflect the results of the pore size distribution calculated using the QSDFT method.

Changes made:

- We have revised Supplementary Fig. 8 and sentence in revised manuscript.

Supplementary Figure 8. Pore size distribution graphs of (e) MOF-808 and (f) MOF-808-EDTA-Cu.

→ (Line 315) "Pore size distribution was calculated using quenched solid density functional theory (QSDFT) method."

3. I noticed that the figures in the main text contain no chemical structure. The structures of important chemicals such as MOF-808, EDTA, and $[Cu(edta)(OH_2)]$ would better be depicted to help the readers understand the system.

We appreciate the reviewer's valuable suggestions about chemical structures for helping the readers understand the system. In response to the reviewer's recommendations, we have added the chemical structures of MOF-808, EDTA, and $[Cu(EDTA)(OH_2)]$ in the revised manuscript and revised supplementary information, as follows.

Changes made:

- We have added the synthesis scheme at Supplementary Figure 1 and revised caption of

Supplementary Figure 6 and the sentences as below.

→ (Line 47) “Chelating ligands such as ethylenediaminetetraacetic acid (EDTA, $C_{10}H_{16}N_2O_8$) ~.”

→ (Line 60) “MOF-808, a robust and porous Zr-based MOF with the ideal structural formula $Zr_6O_4(OH)_4(1,3,5\text{-benzenetricarboxylic acid})_2(HCOO)_6$, was selected as the platform, and Cu-chelated EDTA (Cu-EDTA) was incorporated as the proton-triggered colorimetric center (Supplementary Fig. 1).”

Supplementary Figure 1. Synthesis scheme of MOF-808-EDTA-Cu. BTC is an abbreviation for 1,3,5-benzenetricarboxylic acid.

Supplementary Figure 6. (a) Diffuse reflectance UV-vis-NIR spectra of MOF-808-EDTA-Cu (blue), [Cu(EDTA)(H₂O)] single crystal (green), and MOF-808-EDTA (red). For MOF-808-EDTA-Cu (blue) and [Cu(EDTA)(H₂O)] single crystal (green), the peak appeared at ca. 13140 cm⁻¹ is corresponding to d-d transition of an octahedral six-coordinated of Cu ion. (b) Crystal structure of [Cu(EDTA)(H₂O)], i.e., [Cu(C₁₀H₁₄N₂O₈)(H₂O)], with six-coordinated geometry of Cu as a ball-and-stick model¹.